# Quantum rotor in a two-dimensional mesoscopic Bose gas

**Michał Suchorowski[1][⋆], Alina Badamshina[2], Mikhail Lemeshko[3], Michał Tomza[1], Artem G. Volosniev[3,4][†]**

1 Faculty of Physics, University of Warsaw, Pasteura 5, 02-093 Warsaw, Poland
2 Department of Physics, ETH Zurich, 8093 Zurich, Switzerland
3 Institute of Science and Technology Austria (ISTA), am Campus 1, 3400 Klosterneuburg, Austria
4 Department of Physics and Astronomy, Aarhus University, Ny Munkegade 120, DK-8000 Aarhus C, Denmark

⋆ m.suchorowski@uw.edu.pl , † artem@phys.au.dk

## Abstract

We investigate a molecular quantum rotor in a two-dimensional Bose-Einstein condensate. The focus is on studying the angulon quasiparticle concept in the crossover from few- to many-body physics. To this end, we formulate the problem in real space and solve it with a mean-field approach in the frame co-rotating with the impurity. We show that the system starts to feature angulon characteristics when the size of the bosonic cloud is large enough to screen the rotor. More importantly, we demonstrate the departure from the angulon picture for large system sizes or large angular momenta where the properties of the system are determined by collective excitations of the Bose gas.

# 1   Introduction

The dragging of superfluid helium by a rotating classical object was one of the first manifestations of angular momentum transfer that preserves the superfluidity of helium-4 [1]. After a number of years, similar experiments were performed with quantum rotors – molecules [2]. The surprising conclusion was that a molecule rotates freely within superfluid helium, yet with a different rotational constant [3]. Naturally, it was theorized that some helium atoms must participate in rotation as well [4], just as in the classical analog. A conceptually simple approach for describing the corresponding transfer of the angular momentum from a molecule to the bath is the angulon quasiparticle [5], which captures some of the main features of the available data [6]. However, despite the success of the previous studies of the angulon, their formulation in momentum space does not allow one to address questions concerning superfluidity in small, finite-size systems, which are of both theoretical and experimental interest [7,8].

This paper develops and discusses a framework for studying a quantum rotor immersed in a Bose-Einstein condensate and the ensuing angulon regime in real space (see Fig. 1). The framework is partially based on the idea that a detailed description of boson-boson correlations in the bath is unnecessary, which was one of the key approximations behind the angulon concept. As a result, we can use simple numerical and analytical tools for studying the system instead of Monte Carlo techniques typically employed in analyzing the problem [4]. We also note that our framework allows us to systematically study a departure from the angulon concept, which appears beyond superfluid hydrodynamic models [9], where the microscopic details of boson-boson interactions are also neglected. We illustrate and study the framework for a two-dimensional (2D) system, which received less attention than its three-dimensional counterpart. The 2D quantum rotor in a system of bosons has become accessible experimentally [10], providing additional motivation for our study.

Our real space formulation allows us to go beyond earlier angulon studies in 2D [11] and observe the formation of the angulon by increasing the number of bosons. In this few- to many-body transition, we identify three regimes (see Fig. 1): (i) the few-body regime where the properties of the system are highly sensitive to the number of particles; (ii) the angulon, where the Bose gas screens the impurity, and its effective properties do not change if more bosons are added to the system; (iii) the metastable angulon, where a collective excitation

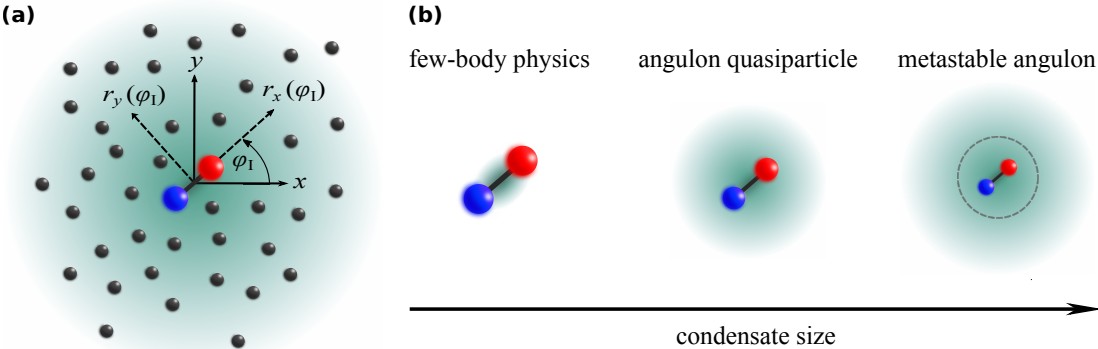

Figure 1: Schematic representation of our study. Panel (a) shows a molecular linear rotor in a two-dimensional system of bosons together with the laboratory $\{x, y\}$ and the molecular $\{r_x, r_y\}$ frames of reference. The latter is determined with respect to the angle $\varphi_I$ that parametrizes rotations of the impurity. The background indicates the condensate size limited by the harmonic trap. Panel (b) illustrates regimes of the system in panel (a) depending on the number of bosons, $N$. For 'small' values of $N$, the rotor and the bosons form a few-body state whose properties depend strongly on $N$. By increasing $N$, the system enters an angulon regime – a quasi-particle that can be described as a dressed rotor. For very large system sizes $N \to \infty$, the angulon becomes unstable, as it can transfer its angular momentum to the bath by exciting collective modes (see the text for details). The gray dashed circle denotes the region directly affected by the impurity.

of the Bose gas defines the lowest energy state of the system for a finite angular momentum. The existence of the regimes (i) and (ii) can be anticipated from Monte Carlo studies in 3D; see, e.g., Ref. [12]. The regime (iii) has been less explored to the best of our knowledge and will be the subject of our forthcoming publication. The focus of this paper is on the angulon regime. We derive an analytical expression for the renormalization of the rotational constant and benchmark it against numerical calculations. Further, we discuss conditions for a breakdown of the angulon concept by increasing the total angular momentum.

The paper is structured as follows. Section 2 describes the theoretical model and our method of solving it. Section 3 presents and discusses the energies as we increase the number of particles, i.e., across a few-to-many-body transition. Section 4 focuses on the properties of the system in the angulon regime. Section 5 summarizes the paper and discusses the outlook for further research. In addition, we include five appendices with technical details of our study.

## 2 Theoretical model and methods

We study an impurity ('molecule') interacting with $N$ bosons in two spatial dimensions; see Fig. 1 (a). The impurity here is a linear rotor with frozen translational degrees of freedom. As our focus is on the physics of the system in the crossover between few- and many-body physics, see Fig. 1 (b), we assume the system is in the harmonic trap, the frequency of which is decreased once we increase the number of particles so that the central density of the Bose gas is fixed. In this section, we formulate the problem and outline our method of solving it.

## 2.1 Hamiltonian

The Hamiltonian of the system reads as

$$\mathcal{H} = \mathcal{H}_{\text{bos}} + \mathcal{H}_{\text{mol}} + \mathcal{H}_{\text{mol-bos}}. \tag{1}$$

Here, $\mathcal{H}_{\text{bos}}$ and $\mathcal{H}_{\text{mol}}$ describe the Bose gas and the molecule, respectively:

$$\mathcal{H}_{\text{bos}} = -\sum_{i=1}^{N} \frac{\hbar^2}{2m_b} \frac{\partial^2}{\partial x_i^2} + \sum_{i=1}^{N} \frac{k|x_i|^2}{2} + \sum_{i<j}^{N} g\delta(x_i - x_j), \qquad \mathcal{H}_{\text{mol}} = -B_{\text{rot}} \frac{\partial^2}{\partial \varphi_{\text{I}}^2}, \tag{2}$$

with $m_b$ being the mass of a single boson, $x_i$ – the coordinate of the $i$-th boson, $k = m_b\omega^2$ – the force constant of the trapping potential, and $g$ – the strength of the boson-boson interaction; $B_{\text{rot}}$ is the rotational constant, and $\varphi_{\text{I}}$ is the corresponding angular coordinate. The sole purpose of boson-boson interactions in our work is to introduce phononic excitations of the Bose gas. Therefore, the specific shape of these interactions is not important for our study, and we parametrize them with Dirac's delta function, $g\delta(x_i - x_j)$. It is well known that this potential is not well-defined and must be regularized by connecting $g$ to physical observables; see, for example, Ref. [13] for a pedagogical discussion. As we shall use the mean-field approximation, the parameter $g$ can, in principle, be written from the two-body physics using the scattering length expressed in the units of the average density [14]. However, in what follows, we choose to relate $g$ to many-body observables, such as the healing length (see below) and the density, which are then used for presenting our findings.

Finally, the impurity-boson interaction has the form

$$\mathcal{H}_{\text{mol-bos}} = \sum_{i=1}^{N} \alpha W(|x_i|, \varphi_i - \varphi_{\text{I}}), \tag{3}$$

where $\alpha$ defines the strength of the interaction; a dimensionless function $W$ describes its shape. In general, one can write $W(x, \varphi) = \sum_m R_m(x)e^{im\varphi}$. As the spherical part, $R_0$, does not lead to any exchange of the angular momentum, we do not consider it here. Instead, we focus on the simplest term that allows for such exchange

$$W(|x_i|, \varphi_i - \varphi_{\text{I}}) = R(|x_i|)\cos(\varphi_i - \varphi_{\text{I}}), \tag{4}$$

where $R(|x_i|)$ describes the radial part of $W$. In our numerical calculations, we will use the potential in the form $R(r) = \frac{r}{r_0}e^{-r^2/r_0^2}$, where $r_0$ determines the range of the potential.

## 2.2 Gross-Pitaevskii equation in the co-rotating frame

As the system is rotationally invariant, the total angular momentum of the system, $L$, is conserved, and any eigenstate of the Hamiltonian can be written as $e^{i\varphi_{\text{I}}L}\psi(r_1, \ldots, r_N)/\sqrt{2\pi}$, where $r_i = (|x_i|, \varphi_i - \varphi_{\text{I}})$ are the coordinates of the bosons measured with respect to the orientation of the impurity, see Fig. 1 (a). It is worthwhile noting that our approach to solving the problem in the co-rotating-with-the-impurity frame of reference using standard approximations of many-body physics is natural for the angulon problem where the particles in the medium are expected to 'adiabatically follow' the rotation of the impurity [4]. At the same time, our results are not applicable when this 'following' does not happen, e.g., if $\alpha = 0$.

The normalized function $\psi$ defines the probability of finding a boson at a given position in a molecular frame of reference. It does not depend explicitly on the angle $\varphi_{\text{I}}$, simplifying

calculations (see Ref. [15] for a relevant discussion of a similar transformation for a three-dimensional problem in the momentum space). We assume that the bosons are weakly interacting and approximate $\psi$ with the product ansatz

$$\psi = \prod_{i=1}^{N} f(r_i). \tag{5}$$

The function $f(r)$ that minimizes the energy of the Hamiltonian (1) must satisfy the Gross-Pitaevskii equation (GPE) in the co-rotating frame

$$\left[ 2iB_{\text{rot}}J\frac{\partial}{\partial\varphi} - B_{\text{rot}}\frac{\partial^2}{\partial\varphi^2} - \frac{\hbar^2}{2m_b}\frac{\partial^2}{\partial r^2} + (N-1)g|f(r)|^2 + \frac{k|r|^2}{2} + \alpha W(r) \right] f = \tilde{\mu}f, \tag{6}$$

where $\tilde{\mu} = \mu - B_{\text{rot}}L^2/N$, and $\mu$ is the chemical potential; $J = L - A$ is the angular momentum of the impurity defined via the quantity

$$A = -i(N-1) \int f^*(r)\frac{\partial}{\partial\varphi}f(r)r\mathrm{d}r\mathrm{d}\varphi, \tag{7}$$

which is the angular momentum of the bath for large $N$. Note that for convenience, we have defined $J$, $A$, and $L$ to be dimensionless. The physical angular momenta are then obtained by multiplying with $\hbar$. The details of the applied transformation and derivation of the GPE can be found in App. A. The total energy of the system is given by

$$E = N\mu - B_{\text{rot}}A^2 - \frac{1}{2}gN(N-1)\int |f(r)|^4\mathrm{d}r. \tag{8}$$

As we are interested in the rotational spectrum of the system, we define the rotational energy $\Delta E_L = E(L) - E(L = 0)$, which will be one of the main subjects of our study.

It is worth noting that our study shares many similarities with the problem of an atom interacting with a finite number of bosons in a ring geometry [16–18]. In particular, the total angular momentum in both cases can be treated as a discrete parameter. This allows us to adopt ideas and methods used in those studies in our work. For example, the effective mass in the Bose polaron problem can be defined from the 'momentum' of the impurity (for a pedagogical introduction, see [19,20]). Similarly, we can use here the Hellmann-Feynman theorem (see App. A) to define the effective rotational constant, $B_{\text{eff}}$, in the limit $L \to 0$ and to connect it to $J$:

$$\Delta E_L = B_{\text{rot}}JL \equiv B_{\text{eff}}L^2, \qquad \text{where} \qquad B_{\text{eff}} = B_{\text{rot}}\frac{J}{L}. \tag{9}$$

$B_{\text{eff}}$ is the parameter that defines the excitation spectrum of the angulon. Although this expression is derived from the assumption of vanishing $L$, we will illustrate below (see Sec. 3.3) that it is also useful for finite values of $L$. Finally, we remark that despite many similarities to the Bose polaron problem, there are also important differences that will become apparent in the course of our study. For example, a two-dimensional Bose angulon is not the system's ground state with $L \neq 0$ in the thermodynamic limit.

## 2.3 Computational details

Here, we provide some general information needed to present our results.

*Units.* We study the system in units where $m_b = \hbar = 1$. To represent our findings in a dimensionless form, we shall use a healing length, which is a natural quantity to describe

condensate properties in the vicinity of the impurity, as it defines the distance over which the Bose gas 'looses' the information about the presence of the impurity. It is defined via the equation $\frac{\hbar^2}{2m_b\xi^2} = ng$, where $n := n(0)$ is the density in the middle of the trap without the impurity. We estimate $n$ using the Thomas-Fermi (TF) approximation[1] [21], assuming that $g$ is independent of the density (cf. Ref. [14]). This leads to $2gn(r) = 2\mu - kr^2$ and $\mu = kR_{\text{TF}}^2/(2g)$, where $R_{\text{TF}} = (4gN/(\pi k))^{1/4}$ is the Thomas-Fermi radius. Therefore, $n = \sqrt{kN/(\pi g)}$. Using the healing length, we can express $B_{\text{rot}}$ and $\alpha$ in the units of $\frac{\hbar^2}{m_b\xi^2}$; $r$ in the units of $\xi$; $k$ in the units of $\frac{\hbar^2}{m_b\xi^4}$; $g$ in the units of $\frac{\hbar^2}{m_b}$. These units are used everywhere except Sec. 3.1, where the range of the potential $r_0$ provides a relevant length scale.

*Parameters for numerical simulations.* Our work aims to study a basic model of a rotating impurity in a two-dimensional Bose gas, focusing on the energies and angular momenta for different numbers of bosons. However, our model is also of experimental interest. For example, a relevant set-up is a molecule on the surface of the superfluid helium nanodroplet [10]. Even though superfluid helium is a strongly correlated system and cannot be described fully within the mean-field approximation, the success of the angulon model in explaining experimental data [6] teaches us that the inclusion of strong correlations within the bath might not be necessary for gaining a qualitative understanding of the rotational energy spectra. Motivated by this observation, we adjust the parameters of our model to mimic some properties of a molecule in helium.

We assume that the healing length is $\xi = 1\,\text{nm}$, similar to the healing length of liquid helium [22, 23]. The bulk density of the three-dimensional liquid helium is $\simeq 22/\text{nm}^3$ (see, e.g., Ref. [24]). Therefore, in our calculations, the strength of boson-boson interactions is set such that $n = n_0 \simeq 10/\xi^2$. Small changes of the parameters leave the qualitative behavior of our results intact, as we illustrate in Sec. 3.3. The value of the rotational constant in our model is around 1 GHz, representing actual values of diatomic molecules [25]. The range of the impurity-bath interactions, $r_0$, in Eq. (4) is set to $\sqrt{2}\xi$, which gives a potential with a minimum at a distance of a few nanometers and the depth of the order of a hundred cm$^{-1}$, which is a reasonable interaction potential [26].

*Numerical method.* To solve the Gross-Pitaevski equation, we use the imaginary time evolution employing the semi-implicit backward Euler scheme. Numerical results can be reproduced using our open-source code available on GitLab [27]. More details can be found in App. B.

## 3 From a one- to a many-boson system

The real space formulation of Eq. (6) is exact for $N = 1$. Furthermore, we expect the mean-field approximation to be accurate for $N \to \infty$ and weak interactions. We start the presentation of our results by considering these limiting cases.

### 3.1 Two-body problem

An analysis of a two-body (one impurity plus one boson) system with $k = 0$ allows us to explore binding between the impurity and a boson. First, we demonstrate that such a system always has at least one bound state extending the known result [28] to $B_{\text{rot}} \neq 0$. To this end, we solve the system in the limit $\alpha \to 0$ following Refs. [29, 30], see App. C for details. Namely, we first

---

[1]We check that using the Thomas-Fermi approximation does not affect our main results aposteriori.

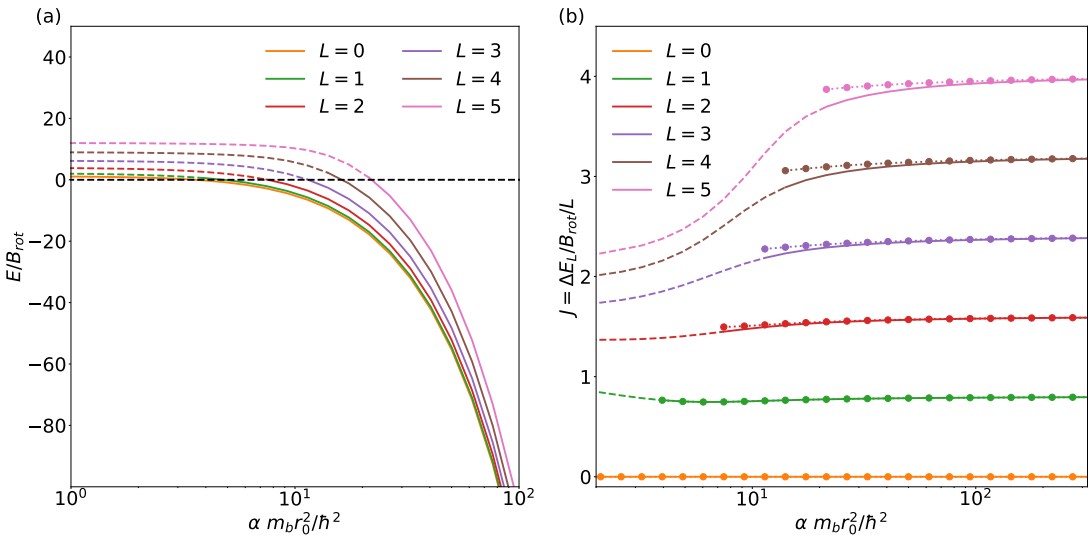

Figure 2: Properties of the two-body system. (a) Energy, $E$, and (b) the angular momentum of the impurity, $J$, defined via Eq. (9) as a function of the interaction strength for different values of the total angular momentum $L$. The dashed curves show the results affected by finite-size effects. The dotted curves with circle markers in panel (b) show $J$ found by fitting effective rotational constant $B_{\text{eff}}$ to the energy in Eq. (9).

expand the wave function in partial waves

$$f(\tilde{r}) = \sqrt{\frac{r_0}{x}} \sum_{m=-1}^{m=1} a_m f_m(\tilde{x}) e^{im\varphi}, \tag{10}$$

where $\tilde{r} = (\tilde{x}, \varphi)$ is a dimensionless coordinate with $\tilde{x} = x/r_0{}^2$. Then, we solve the Schrödinger equation perturbatively. Note that the expansion in Eq. (10) is limited to $|m| = 0, 1$ because the considered potential can couple only these states in the lowest order in $\alpha$. It is straightforward to extend the derivation to other potentials.

We find that the system is always bound for $L = 0$. The threshold behavior (the limit $\alpha \to 0$) of the ground-state energy is

$$E \propto -\exp\left(-2\frac{\hbar^4 r_0^4}{m_b^2 \alpha^2 \kappa_0}\right), \tag{11}$$

where $\kappa_0 > 0$ is a coefficient determined by the shape of the potential and $B_{\text{rot}}$, see App. C for more details. The value of $\kappa_0$ decreases when $B_{\text{rot}}$ increases. Correspondingly, the system becomes less bound, in agreement with the expectation that the kinetic energy of the impurity can only increase the energy. Further, we find that weakly interacting systems with $|L| > 0$ are not bound.

To investigate the system beyond the $\alpha \to 0$ limit, we perform numerical calculations in a finite box. Note that there are severe finite-size effects for weak interactions. For example, the size of the ground state is expected to be exponentially large $\propto \exp\big(\hbar^4 r_0^4/(m_b^2 \alpha^2 \kappa_0)\big)$ (cf. Ref. [29]), which poses a difficulty for any numerical technique in a finite box. Therefore,

---

[2]In this subsection, we use $r_0$ as the unit of length because the healing length – the standard unit of length in our work – cannot be defined for a two-body problem.

here we focus only on the interactions for which the system is 'sufficiently' bound so that its properties are weakly affected by the boundary conditions. To estimate the value of $\alpha$ at the threshold for binding for $L \neq 0$, we note that the lowest-energy state for a non-interacting system for every value of $L$ describes a non-rotating impurity and a boson that carries the angular momentum. This state has a vanishing energy, providing a reference point for binding.

We demonstrate our numerical findings in Fig. 2 for $B_{\rm rot} = 0.125\, m_b r_0^2/(2\hbar^2)$. Finite-size effects alter results for small values of $\alpha$. However, as soon as $E < 0$ our results quickly become accurate. Note that the system is less bound when $L$ increases in agreement with our analytical derivations above. It is also worthwhile noting that the energy spectrum of the system follows $B_{\rm eff}L^2$ for $\alpha \to \infty$, where $B_{\rm eff} \simeq 0.8 B_{\rm rot}$ is an effective rotational constant. This is expected as the boson attaches to the impurity, decreasing the rotational constant. Indeed, simple analytical calculations based upon the idea that the impurity can only move close to the minimum of the impurity-boson potential $\left(r = r_{\rm min} = \frac{r_0}{\sqrt{2}}\right)$ lead to the renormalization of the effective constant

$$\frac{B_{\rm eff}(\alpha \to \infty)}{B_{\rm rot}} = \frac{1}{1 + \frac{2IB_{\rm rot}}{\hbar^2}} \qquad \to \qquad \frac{B_{\rm eff}(\alpha \to \infty)}{B_{\rm rot}} = 0.8, \tag{12}$$

in excellent agreement with our numerical estimate. Here, $I = m_b r_{\rm min}^2$ is the classical moment of inertia of the boson.

## 3.2 Angulon in the thermodynamic limit

Here, we derive an approximate analytical solution of Eq. (6) in the limit $k \to 0$ (assuming a fixed density of the Bose gas far from the impurity) and $\alpha \to 0$, which provides insight into general properties of the angulon quasiparticle. To this end, we consider the system with $J \neq 0$ and write the corresponding function $f$ as

$$f(\mathbf{r}) = f_0(r) + f_{\rm r}(\mathbf{r}) + i f_{\rm i}(\mathbf{r}), \tag{13}$$

where $f_0(r) = \sqrt{\frac{n}{N}}$ is the Thomas-Fermi profile in the limit $k \to 0$. The density of the condensate without the impurity, $n$, is constant by assumption. To find $f_{\rm r}$ and $f_{\rm i}$, we first linearize the GPE with $\alpha$ as a small parameter. Then, we solve the resulting equations in an integral form (see App. D). We find

$$f_{\rm r}(\mathbf{r}) = -2\frac{m_b}{\hbar^2}\alpha\sqrt{\frac{n}{N}}F_{\rm r}(r)\cos\varphi, \qquad f_{\rm i}(\mathbf{r}) = -8\frac{m_b^2}{\hbar^4}B_{\rm rot}J\alpha\sqrt{\frac{n}{N}}F_{\rm i}(r)\sin\varphi, \tag{14}$$

where the radial parts have the form

$$F_{\rm r}(r) = I_1(br)\int_r^\infty K_1(br')R(r')r'{\rm d}r' + K_1(br)\int_0^r I_1(br')R(r')r'{\rm d}r', \tag{15}$$

and

$$F_{\rm i}(r) = I_1(cr)\int_r^\infty K_1(cr')F_{\rm r}(r')r'{\rm d}r' + K_1(cr)\int_0^r I_1(cr')F_{\rm r}(r')r'{\rm d}r'. \tag{16}$$

Here, $b = \sqrt{\frac{m_b}{\hbar^2}2B_{\rm rot}}$, $c = \sqrt{\frac{m_b}{\hbar^2}(2B_{\rm rot} + 4gn)}$ and $I_1(r)$, $K_1(r)$ are modified Bessel functions of the first and second kind, respectively [31]. As it will become evident in the next section, these expressions do not correspond to the system's ground state. Instead, they describe the angulon solution.

Using the above solution, we can find an expression for the angular momenta of the bath and the impurity

$$A = \frac{32\pi\frac{m_b^3}{\hbar^6}nB_{\rm rot}\alpha^2 H_{\rm ri}}{1 + 32\pi\frac{m_b^3}{\hbar^6}nB_{\rm rot}\alpha^2 H_{\rm ri}}L, \qquad J = \frac{1}{1 + 32\pi\frac{m_b^3}{\hbar^6}nB_{\rm rot}\alpha^2 H_{\rm ri}}L, \qquad (17)$$

where $H_{\rm ri} = \int_0^\infty {\rm d}r F_{\rm r}(r)F_{\rm i}(r)r$ describes the overlap between the real and imaginary parts of the order parameter. Note that there is no linear dependence on $\alpha$ in Eq. (17) because of our choice of the impurity-boson potential. Indeed, $\alpha \to -\alpha$ cannot change the physics of our set-up.

The effective rotational constant can now be easily found

$$B_{\rm eff} = \frac{1}{1 + 32\pi\frac{m_b^3}{\hbar^6}nB_{\rm rot}\alpha^2 H_{\rm ri}}B_{\rm rot}. \qquad (18)$$

By comparing Eqs. (12) and (18), we can conclude that the parameter $16\pi m_b^3 n\alpha^2 H_{\rm ri}/\hbar^4$ is the moment of inertia of the bosons that adiabatically follow the impurity. Equation (18) is derived in the assumption of small values of $\alpha$, and therefore, strictly speaking, we should rewrite it as $B_{\rm eff} - B_{\rm rot} \simeq -32\pi\frac{m_b^3}{\hbar^6}nB_{\rm rot}^2\alpha^2 H_{\rm ri}$. However, we will show numerically that, in fact, Eq. (18) leads to qualitatively correct results for all $\alpha$. For large values of $\alpha$, i.e., $\alpha \to \infty$, $B_{\rm eff}$ vanishes. This is expected as, in this case, many bosons can be attached to the rotor. In other words, increasing $\alpha$ transfers the angular momentum from the molecule to the bath.

The effective parameter $B_{\rm eff}$ cannot differ significantly from $B_{\rm rot}$ for very low condensate densities or very weak interactions, in agreement with previous calculations [32]. One should have a sufficiently dense medium and strong interactions to observe the renormalization of impurity parameters. The interplay between the density of the Bose gas and the strength of the interaction manifests itself in the fact that $nB_{\rm rot}\alpha^2 H_{\rm ri}$ is the only parameter in Eq. (18) that contains microscopic information about the system.

## 3.3   Transition between the limits

Here, we investigate Eq. (6) numerically in between the limits discussed above, which allows us to understand the formation of the angulon in the bottom-up approach. To this end, we first calculate the energy spectrum as a function of the number of particles, $N$. This is a natural way to understand the connection between the two limits. Then, we consider the spectrum as a function of the system's density, $n$.

*Energy spectrum as a function of $N$.* To set the density in the middle of the trap to $n_0$, we fix $g = 1$ and keep the ratio $kN = 500$ (recall that $n = \sqrt{kN/(\pi g)}$ in the Thomas-Fermi approximation). Other parameters used in the simulations are $B_{\rm rot} = 0.125$ and $r_0 = \sqrt{2}$; Section 2.3 explains the units and motivates this choice of dimensionless parameters.

In Figure 3, we demonstrate the first rotational energy $\Delta E_1$ and the bath angular momentum $A$ for $L = 1$. We show both quantities on purpose to illustrate the applicability of Eq. (9), according to which $A = 1 - \Delta E_1/B_{\rm rot}$. For all values of $\alpha$, we observe three distinct regimes by changing $N$, which we shall unimaginatively call (i) a few-body regime, (ii) an angulon plateau, and (iii) a collective excitation.

In the few-body regime, the physics of the system is dominated by the external trap[3] (the size of the condensate, $R_{\rm TF}$, is smaller than the range of interaction). The Bose gas is forced

---

[3]For example, we do not reproduce results from Sec. 3.1. Instead, for $N = 1$, the molecule rotates as an unperturbed rotor because the excitation energy of the boson (i.e., $\hbar\omega$) is large in comparison to the interaction energy for the considered parameters.

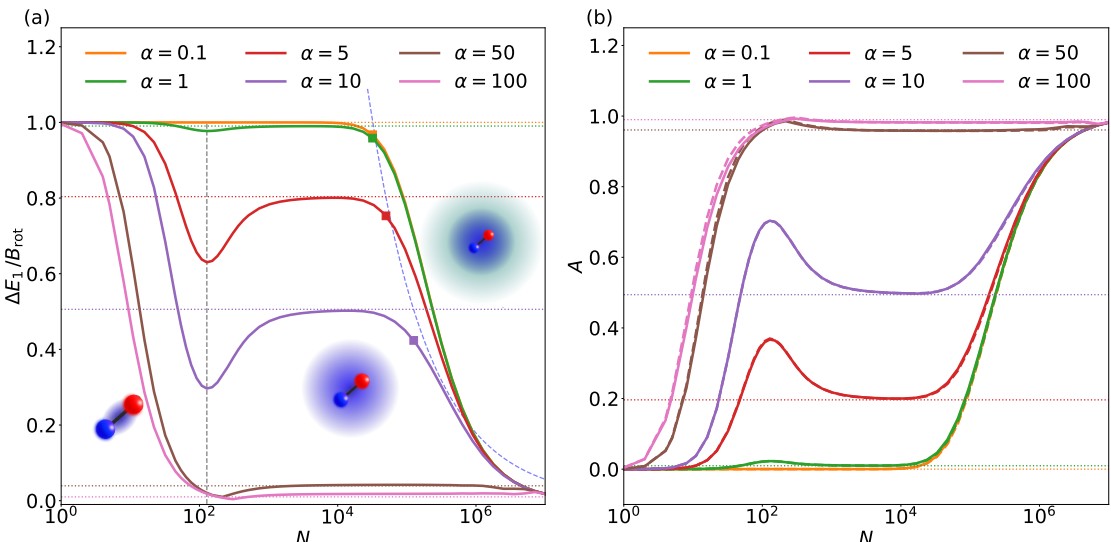

Figure 3: (a) First rotational energy $\Delta E_1$ and (b) bath angular momentum $A$ for $L = 1$ as a function of the number of bosons $N$ for different values of the molecule-bath interaction strength $\alpha$ (shown in the units of $\hbar^2/(m_b\xi^2)$ in every panel). Horizontal dotted lines show the analytical values for given $\alpha$ computed by inserting Eq. (18) into Eq. (9). The black dashed vertical line in (a) shows a minimum of the energy, which signals a transition from a few-body state to an angulon regime. The (blue) dashed curve in (a) denotes the energy $\hbar\omega$ as a function of $N$. Its crossings with horizontal dotted lines provide an estimate for the number of bosons $N$ (size of the system) when collective excitations of the Bose gas become lower in energy than the angulon state. Squares denote the $\Delta E_1$ values at the expected transition point. Dashed curves in panel (b) show the value of $A = 1 - \Delta E_1/B_{\text{rot}}$ (see Eq. (9)), while the solid curves show the result of direct calculations with Eq. (7).

to occupy the region close to the impurity. By increasing $R_{\text{TF}}$, we observe a change in the system's behavior. In particular, we observe numerically a minimum of $\Delta E_1$ when $R_{\text{TF}} \simeq \beta r_0$ with $\beta = 1.8$, see the vertical dashed line in Fig. 3(a). Surprisingly, the value of $\beta$ appears to be fixed only by the value of $R_{\text{TF}}/r_0$. In particular, it is independent of the value of $\alpha$. We conclude that when the condensate size exceeds the interaction range, i.e., when the bosons screen the impurity, we observe a few-body-to-angulon transition. By integrating the radial shape of the potential $R(r)$ from 0 to $R_{\text{TF}}$, we find $\frac{\int_0^{\beta r_0} R(r)r\,dr}{\int_0^\infty R(r)r\,dr} = \text{erf}\,\beta - \frac{2}{\sqrt{\pi}}\beta \exp(-\beta^2) \simeq 0.9$. This shows that the Bose gas occupies most of the potential at the transition point.

After the impurity is screened, the rotational energy is characterized by the *angulon plateau* that is described by the analytical results from Sec. 3.2 well (see the dotted horizontal lines in Fig. 3). In the next section, we discuss this regime and connect it to the angulon concept. As the size of the bath is increased even further, another transition takes place. In a numerical solution in the non-interacting limit ($\alpha \to 0$) for large systems ($N \simeq 10^4 - 10^5$), the lowest energy state for $L = 1$ becomes disconnected from the molecular rotation. We leave an analysis of this transition and the corresponding metastable angulon state to future studies. Let us, however, estimate the transition point by assuming that the lowest energy of the Bose gas with $L = 1$ has the energy $\hbar\omega$, which corresponds to the lowest center-of-mass excitation, i.e., rotation of the Bose gas as a whole[4]. The transition then happens when the energy of the

---

[4]This state and collective excitations of the Bose gas are missing in the analytical solution in Sec. 3.2, which

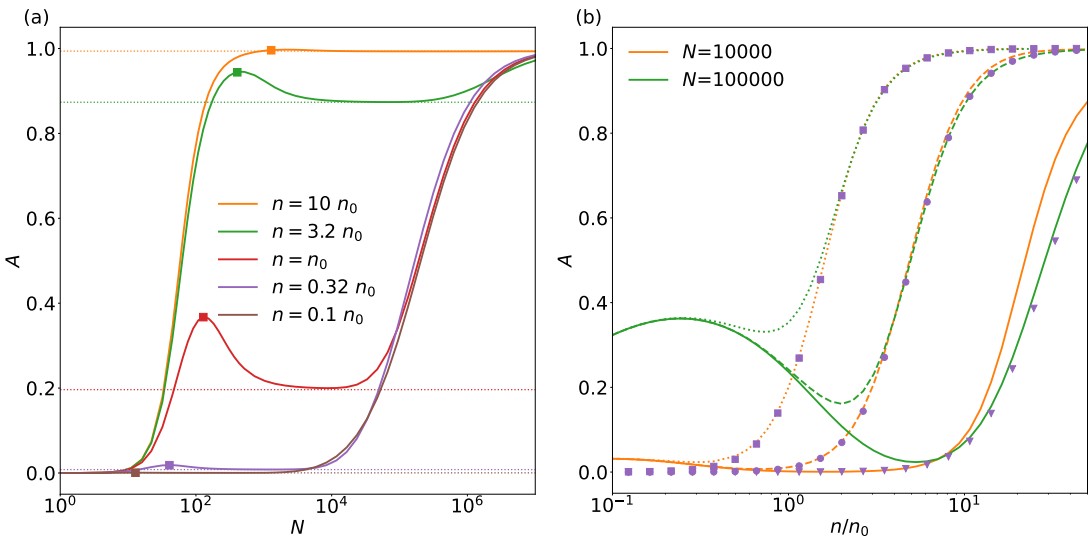

Figure 4: (a) Bath angular momentum $A$ for $L = 1$ as a function of the number of bosons, $N$, for different central densities $n$ and fixed $\alpha = 5\,m_b\xi^2/\hbar^2$. The dotted horizontal lines show Eq. (17). (b) Bath angular momentum $A$ for $L = 1$ as a function of the central density $n$ for different numbers of bosons, $N$. The interaction strength is given by $\alpha = 0.1\,m_b\xi^2/\hbar^2$ (solid), $\alpha = m_b\xi^2/\hbar^2$ (dashed) and $\alpha = 5\,m_b\xi^2/\hbar^2$ (dotted). Violet markers show the analytical results, Eq. (17), for $\alpha = 0.1\,m_b\xi^2/\hbar^2$ (triangles), $\alpha = m_b\xi^2/\hbar^2$ (circles) and $\alpha = 5\,m_b\xi^2/\hbar^2$ (squares).

angulon $E = B_{\mathrm{eff}}L^2$ is larger than $\hbar\omega$. In Fig. 3(a), the former is shown as a dotted horizontal line for different values of $\alpha$; the latter is a blue dashed curve independent of $\alpha$. The values of $N$ for which the two cross are presented as squares. Reasonable agreement between the squares and the departure from the angulon plateau demonstrates that the transition point can be captured using a basic physical picture, at least at a qualitative level.

*Energy spectrum for different densities.* Here, we study how the condensate's density affects the results presented in Fig. 3. For convenience, we use $n_0 = \sqrt{k_0 N_0/(\pi g_0)}$ (here $k_0 N_0 = 500$ and $g_0 = 1$) as a unit of $n$. We change $n$ by adjusting the value of $g$, while keeping $kN = 500$ fixed.

In Figures 4(a) and 4(b), we show the bath's angular momentum as a function of the number of bosons for different central densities and as a function of the density for different numbers of bosons, respectively. Panel (a) demonstrates that changing the density shifts the transition point between the angulon plateau and the collective excitation regimes. In addition, we see that the maximum of $A$ shifts. We interpret that the number of bosons needed to screen the impurity and form the angulon quasiparticle is affected by $n$ (or rather by $g$ used to adjust $n$). To support this interpretation, we note that the condition for the maximum of $A$ postulated above: $R_{\mathrm{TF}} \simeq \beta r_0$ with $\beta = 1.8$, is still accurate (see square markers in Fig. 4(a)). The shift of the maximum of $A$ is driven by the change in the Thomas-Fermi radius $R_{\mathrm{TF}} \propto \sqrt{N}\sqrt[4]{g}$.

By comparing Figs. 4(a) and 3(b), we see that changing density $n$ has a similar effect to the change of $\alpha$ for the properties of the system. In particular, the qualitative behavior of $A$ in the limit $n \to \infty$ ($n \to 0$) appears similar to $\alpha \to \infty$ ($\alpha \to 0$). We conclude that the system's physics is driven by the interplay between impurity-boson interactions and the bath density, in agreement with the analytical results where the key parameter is $n\alpha^2$, see Eq. (17).

As a further remark on this point, note that in Fig. 4(b), the system is effectively non-

---

hence cannot describe the ground state.

interacting in the limit $n \to 0$; the properties of the system no longer depend on the coupling strength. Even though our model is not applicable at low densities, for which 'adiabatic following' of the Bose gas does not happen, the renormalization of the effective rotational constant must indeed be weak in low-density condensates in agreement with the angulon studies in momentum space [11, 32].

Finally, we compare our numerical results with the analytical results of Sec. 3.2. In Fig. 4(a), we observe that the analytical solution (dotted horizontal line) adequately describes the angulon plateau. In Fig. 4(b), we see that the results of Sec. 3.2 agree with the numerical data at high densities. If the density is decreased, the lowest energy state for $L = 1$ is determined by the collective modes of the Bose gas, leading to a difference between markers and curves in Fig. 4(b).

# 4 Angulon regime

## 4.1 Angulon properties from weak to strong interactions

In this section, we focus on the angulon regime and low angular momenta, $L \leq 3$. In particular, we investigate how the angulon properties are affected by the interaction strength $\alpha$. We fix the number of particles, $N = 1000$, and the density, $n_0$, for our investigation. Other parameters are chosen as in the previous section, i.e., $kN = 500$, $B_{\rm rot} = 0.125$, and $r_0 = \sqrt{2}$ in the units of Sec. 2.3. One remark is in order here: in this paper, we focus on the lowest-energy states in each $L$-block and do not calculate the spectral function. Therefore, when we interpret our results in terms of the angulon quasiparticle[5], we actually rely on the previous literature [5, 11].

First of all, we compute the low-energy spectrum (see Fig. 5). We observe that it is described well by $E(L) = E(L = 0) + B_{\rm eff}L^2$ in agreement with the angulon theory [11, 32]. We find the value of the effective rotational constant by fitting the expression $B_{\rm eff}L^2$ to the $\Delta E_L$. In the weak-coupling regime ($\alpha \to 0$), the renormalization of the effective rotational constant is negligible; the 2D free rotor energy, $\Delta E_L \simeq B_{\rm rot}L^2$, determines the spectrum. In this regime, the molecule enjoys having all angular momentum of the system, i.e., $J \simeq L$. For stronger interactions, the molecule becomes dressed, which implies a noticeable renormalization of the rotational constant and the transfer of the angular momentum to the bath. In the strong interaction limit ($\alpha \to \infty$), the bath has most of the angular momentum, i.e., $A \simeq L$. The effective rotational constant tends to zero, wiping out the energy gap between the ground state and the lowest energy level with $L = 1$. Although this behavior is expected from the previous angulon studies in momentum space (see, e.g., Ref. [33]), its systematic investigation is arguably more transparent in real space. Indeed, a strong deformation of the density of the Bose gas requires a number of phonons in momentum space for its description.

For a three-dimensional angulon, strong interactions imply an angular self-localization transition, at least within certain approximations schemes [34]. Our results also show a crossover to a state localized in space. To study this crossover, we study the kinetic and potential energy of the impurity defined as follows

$$E_{\rm I}^{\rm kin} = \left\langle \psi \left| -B_{\rm rot} \frac{\partial^2}{\partial \varphi_{\rm I}^2} \right| \psi \right\rangle = B_{\rm rot}\left(J^2 + \left\langle A^2 \right\rangle\right), \qquad E_{\rm I}^{\rm pot} = \alpha \left\langle \psi | R(r) \cos \varphi | \psi \right\rangle, \qquad (19)$$

where we have introduced $\left\langle A^2 \right\rangle \equiv N \left\langle f | -\frac{\partial^2}{\partial \varphi^2} | f \right\rangle$. These quantities are shown in the inset of Fig. 5(b). As the interaction increases, the molecule localizes in the space of possible angular

---

[5]For example, when we interpret $\Delta E_L$ in the angulon regime as the excitation energy of the quasiparticle.

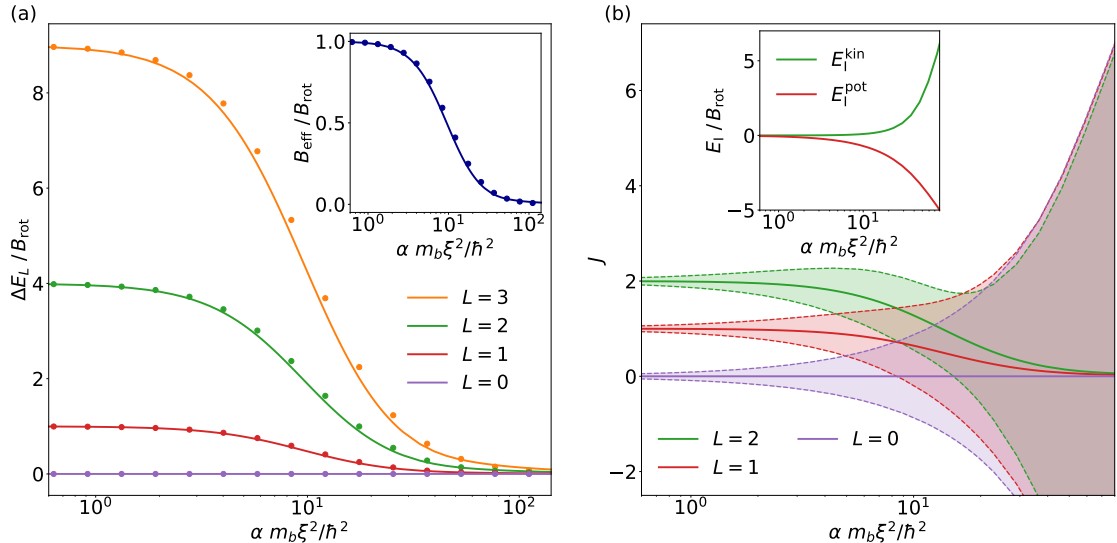

Figure 5: (a) Rotational energy $\Delta E_L$ as a function of the interaction strength $\alpha$. The inset shows the effective rotational constant $B_{\text{eff}}$ as a function of $\alpha$. Dots demonstrate Eq. (9). (b) Angular momentum of the molecule $J$ and its variance $\text{Var}[J]$. The dashed curves and the shaded area illustrate $J \pm \text{Var}[J]$. The inset shows the kinetic (potential) energy of the molecule $E_{\text{I}}^{\text{kin}}\left(E_{\text{I}}^{\text{pot}}\right)$ for $L = 0$.

orientations to minimize the interaction energy. At the same time, the kinetic energy diverges – the state of the impurity becomes delocalized in angular momentum space.

The kinetic energy increase agrees with the uncertainty principle for the angular momentum and orientation [35,36]. Indeed, let us look at the variance of $J$ that gives us a dispersion of the occupations of different angular momentum states. To define it, we again use $\langle A^2 \rangle$ and we note that according to our notation $\langle A \rangle \equiv A = N \langle f | -i \frac{\partial}{\partial \varphi} | f \rangle$ and $\langle J \rangle \equiv J = L - A$. From that follows $\langle J^2 \rangle = \langle A^2 \rangle - 2LA + L^2$ and

$$\text{Var}[J] = \langle J^2 \rangle - J^2 = \langle A^2 \rangle - A^2. \tag{20}$$

In Fig. 5(b), we show that while the expectation value of the $J$ vanishes because of a 'heavy dress' of the rotor, its variance diverges due to the delocalization of the state in angular momentum space.

The vanishing value of $J$ diminishes the usefulness of the angulon concept because the energy spectrum becomes dense. Still, it is worthwhile noting that our numerical results agree with the analytical prediction of Eq. (9) also in the strongly interacting regime, providing us with a simple way to estimate the ground-state energy for each $L \leq 3$-manifold and for every value of $\alpha$. This agreement worsens when we increase the angular momentum $L$, which is expected as the analytical solution was derived in the limit $L \to 0$.

## 4.2   Disintegration of the angulon by increasing the angular momentum

Here, we discuss the system's behavior as a function of the total angular momentum $L$. The goal is to demonstrate that the energy of the system does not follow that of the angulon quasi-particle above a certain angular momentum. We use $N = 5000$, $\alpha = 5$, and $n = n_0$ for this investigation. Other parameters are chosen as in the previous sections, i.e., $kN = 500$, $B_{\text{rot}} = 0.125$, and $r_0 = \sqrt{2}$ in the units of Sec. 2.3.

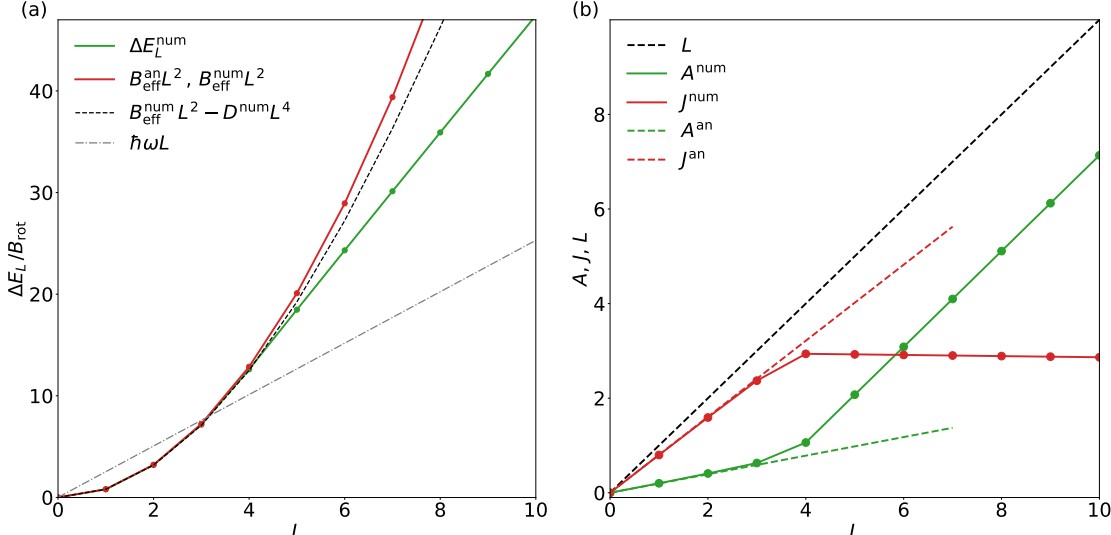

Figure 6: (a) Energy $\Delta E_L$ as a function of the total angular momentum, $L$. The green curve demonstrates the energy calculated numerically; the red curve presents $B_{\text{eff}} L^2$ as predicted by angulon theory. Here, $B_{\text{eff}}^{\text{num}}$ is obtained by fitting to the numerical data – it agrees well with the analytical result, $B_{\text{eff}}^{\text{an}}$ of Sec. 3.2. The black dashed curve is the fit to the angulon energy that includes the centrifugal distortion term, $DL^4$. The gray dash-dotted line shows the energy of bosonic excitations $\hbar\omega L$. (b) Angular momentum of the bath, $A$, the molecule, $J$, and the total, $L$, as functions of $L$. Dashed lines show analytical results of Sec. 3.2; dots are computed numerically. In both panels, solid lines are added to guide the eye.

In Fig. 6(a), we present the energy of the system as a function of the total angular momentum $L$. We see that in the low angular momentum regime ($L \leq 3$), the system is well described by the effective rotational constant $B_{\text{eff}}$, i.e., the energy increases in accordance with $B_{\text{eff}} L^2$. This suggests that one can use the known analytical results (see, e.g., Ref. [37, 38]) for the planar rotor to describe this regime of the many-body system in a simple manner, also in weak external fields. Note also an excellent agreement between the effective rotational constant fitted to numerical results and the findings of Sec. 3.2.

For $L > 3$, we observe a departure from the angulon concept – the energy does not follow the simple $B_{\text{eff}} L^2$ law. We can slightly improve the angulon framework in this region by adding to the energy a centrifugal distortion term $DL^4$, which was successfully used before to take into account the non-rigidity of the angulon quasiparticle [39]. However, in our study, this term leads only to a slight improvement, indicating that the breakdown of the angulon picture is not connected to centrifugal distortion.

Further, we calculate the angular momentum of the molecule, $J$, see Fig. 6(b). We see that for $L > 4$ (i.e., when the angulon picture does not describe the energy anymore) we no longer see an increase in $J$. We conclude that there exists a critical angular momentum $L_{\text{crit}}$ when the excitations of the bath become more favorable than the excitation of the molecule. This resembles a translational motion in a superfluid above Landau's critical velocity. To estimate $L_{\text{crit}}$, we assume that the lowest energy excitation of the Bose gas for a given $L$ is of the order of $\hbar\omega L$, thus extending the discussion in Sec. 3.3. Using this estimate, we conclude that for the considered system $L_{\text{crit}} \simeq 3$, see Fig. 6(a). This value is in agreement with our numerical analysis. Thus, the critical rotation of the angulon quasiparticle can indeed be estimated using a simple expression $L_{\text{crit}} \simeq \hbar\omega/B_{\text{eff}}$.

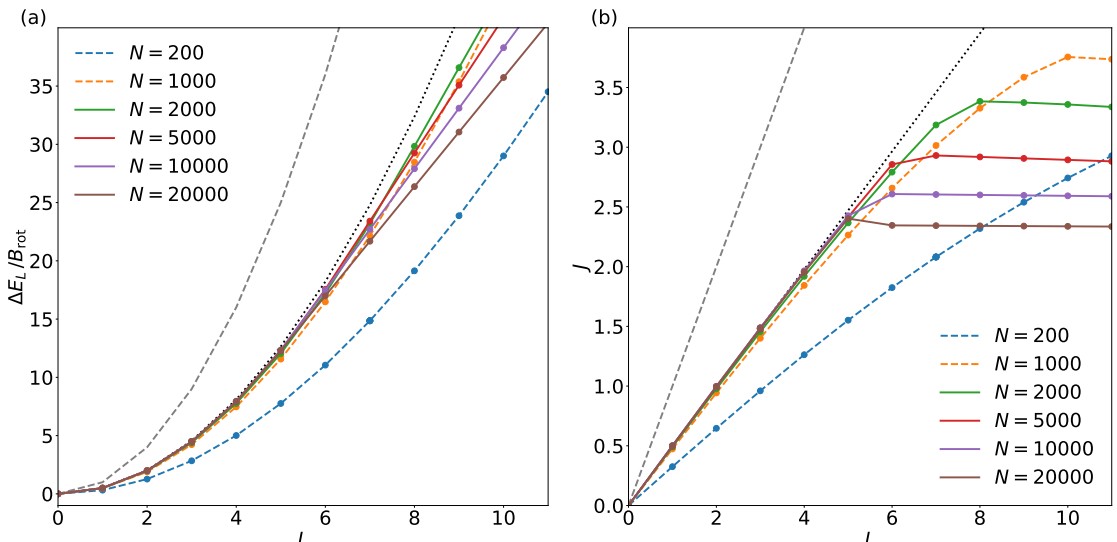

Figure 7: (a) Rotational energy, $\Delta E_L$, and (b) the angular momentum of the molecule, $J$, as functions of the total angular momentum $L$ for different sizes of the condensate, $N$. The gray dashed curves show the results for the free rotor. The black dotted curves are the analytical results of Sec. 3.2. Markers present our numerical results; lines between the markers are added to guide the eye. Note that the corresponding results in the few-body limit can be found in App. E (see Fig. 11). In this limit, we do not have critical angular momenta because the harmonic trap dominates the physics of the system.

This expression suggests that the critical rotation velocity strongly depends on the size of the condensate. To illustrate this, we perform a numerical analysis of a system with $\alpha = 10$ and the density of the Bose gas $n_0$, see Fig. 7. Other parameters are as before, i.e., $kN = 500$, $B_{\text{rot}} = 0.125$, and $r_0 = \sqrt{2}$ in the units of Sec. 2.3. For $N \simeq 200$, the system is in between the few-body and the angulon regimes. The harmonic trap does not dominate the system properties altogether, still the system is too small to obey the angulon physics, cf. Fig. 3. For $N \gtrsim 1000$, the system follows the angulon predictions. In particular, the rotational energy and the angular momentum agree with the analytical calculations (dotted curves) in the low angular momentum region, $L \leq L_{\text{crit}}$.

As expected, the value of the critical angular momentum, $L_{\text{crit}}$, is affected by the size of the condensate. For larger condensates, the critical angular momentum is smaller, which agrees with the observation that the departure from the angulon picture is driven by collective excitations of the Bose gas so that $L_{\text{crit}} \simeq \hbar\omega/B_{\text{eff}}$. Further, we observe that this condition works best in the middle of the angulon plateau (see Fig. 3). In particular, it predicts that $L_{\text{crit}} = 12, 8, 5, 3, 2$ for $N = 1000, 2000, 5000, 10000, 20000$, respectively. The numerically calculated values are in reasonable agreement with this prediction, $L_{\text{crit}} = 9, 7, 5, 5, 5$.

## 5   Summary and Outlook

In this work, we have studied a quantum rotor in a 2D bosonic condensate. We investigated the effect of the size of the condensate, its density, and the coupling strength between the molecule and the bath on the rotational energy of the system, its angular momentum distribution, and the renormalization of the rotational constant.

## 5.1  Summary

This paper can be summarized as follows:

- We proposed to use a Gross-Pitaevskii-like equation to model a molecular impurity coupled to a 2D bosonic bath. This real-space formalism allowed us to analyze finite size effects in the context of the angulon concept and to identify three parameter regimes with respect to the condensate size: the few-body regime, which physics is mainly driven by the external harmonic trap; the angulon regime, where the system is well described by the angulon quasiparticle picture; and the many-body limit where the collective excitations of the Bose gas dominate the low-energy dynamics of the system. This implies that the angulon quasiparticle is a metastable state in the thermodynamic limit.

- We analyzed analytically the GPE without a trap in the two limiting cases: for a single boson and for a system in the thermodynamic limit. We showed that a former set-up always supports at least one bound state. For weak interactions this bound state is formed only for vanishing angular momentum. By increasing interactions, bound states can also exist with non-zero angular momenta. An analytical solution derived in the thermodynamic limit was used as a benchmark for our numerical analysis, summarized below.

- We analyzed the GPE numerically to understand the properties of the system for general parameters. We observed that a *dilute* bath is weakly coupled to the impurity and that the corresponding ground state is determined by a collective mode of the Bose gas. When the condensate is *dense*, the rotation of the molecule 'slows down' by transferring angular momentum to the bath, which renormalizes the effective rotational constant of the molecule. Further, our numerical analysis showed that the angulon only forms below the critical angular momentum, which can be estimated in a simple manner (cf. Landau's critical velocity).

## 5.2  Outlook

Let us briefly identify a few open questions motivated by our study. The primary limitation of our model lies in its mean-field approach, which may not fully capture the intricate properties of strongly correlated helium. In this light, we regard our model as a minimal representation whose predictions are expected to hold because previous studies on angulons managed to explain certain aspects of experimental data concerning helium nanodroplets [6]. Still, a more careful analysis of the beyond-mean-field effects is needed to understand the limits of validity for the results presented in this work.

Further, we plan to investigate the metastable angulon regime, in particular, the excitations of the Bose gas that define the corresponding ground state of the system. It appears unavoidable to compare and contrast this regime of a finite system with angulon instabilities that were predicted in the thermodynamic limit [32] and later employed to explain anomalous broadening of the spectroscopic lines of $CH_3$ and $NH_3$ in helium nanodroplets [40]. We are also interested in the limit of 'very' large angular momenta that can lead to the emergence of non-linear excitations (such as vortices) in helium nanodroplets. It is particularly interesting to extend these studies to the 3D geometries, which are of significant experimental interest.

Finally, we note that the considered impurity-boson potential always supports a bound state because $\int W(x, \varphi) x \, \mathrm{d}x \, \mathrm{d}\varphi = 0$. It is worthwhile considering potentials that do not support a bound state in 2D, i.e., the potentials with $\int W(x, \varphi) x \, \mathrm{d}x \, \mathrm{d}\varphi > 0$. We leave a corresponding investigation to further studies, which might be especially interesting close to the threshold for binding where one might expect a resonant angular momentum transfer; see the discussion

for a three-dimensional problem in Ref. [33]. In general, the potentials with a strong spherical component should be used for describing the realistic impurity-boson interactions [15, 41].

## Acknowledgements

We thank Fabian Brauneis, Arthur Christianen and Pietro Massignan for useful discussions.

**Funding information**    M. S. and A. G. V. would like to thank the Institut Henri Poincaré (UAR 839 CNRS-Sorbonne Université) and the LabEx CARMIN (ANR-10-LABX-59-01) for their support and hospitality during the final stages of completion of this work. M.S. and M.T. acknowledge the National Science Centre, Poland, within Sonata Bis Grant No. 2020/38/E/ST2/00564. M.L. acknowledges support by the European Research Council (ERC) Starting Grant No.801770 (ANGULON). M.S. acknowledges the National Science Centre, Poland, within Preludium Grant No. 2023/49/N/ST2/03820. We gratefully acknowledge Poland's high-performance Infrastructure PLGrid ACK Cyfronet AGH for providing computer facilities and support within computational grant no PLG/2023/016878.

## A    Derivation of Gross-Pitaevskii equation in co-rotating frame

To derive Eq. (6), firstly, we introduce a dimensionless unit of length $x_i/\xi$, where $\xi$ is the healing length of the condensate. The Hamiltonian then takes the form

$$\mathcal{H} = -\sum_i^N \frac{\hbar^2}{2m_b\xi^2}\frac{\partial^2}{\partial(x_i/\xi)^2} - B_{\text{rot}}\frac{\partial^2}{\partial\varphi_{\text{I}}^2} + \sum_{i<j}^N g\frac{1}{\xi^2}\delta(|x_i - x_j|/\xi)$$
$$+ \sum_i^N \alpha W(x_i/\xi, \varphi_i - \varphi_{\text{I}}) + \sum_i^N \xi^2\frac{k|x_i/\xi|^2}{2} \equiv \frac{\hbar^2}{m_b\xi^2}\tilde{\mathcal{H}}, \qquad (A.1)$$

where $\tilde{\mathcal{H}}$ is a dimensionless Hamiltonian and the many-body wavefunction is

$$\Psi(\{x_i\}, \varphi_{\text{I}}) = \frac{1}{\xi^N}\tilde{\Psi}(\{x_i/\xi\}, \varphi_{\text{I}}). \qquad (A.2)$$

We work with the basis set

$$\left\{R_{n_1,\dots,m}(\{|x_i|/\xi\})e^{in_1\varphi_1 + in_2\varphi_2 + \dots + in_N\varphi_N}e^{im\varphi_{\text{I}}}\right\}, \qquad (A.3)$$

where $n_i$ and $m$ are integers that define the angular momentum for each particle in a non-interacting system; $R_{n_1,\dots,m}$ is the radial part of the wave function. As the interaction depends only on the relative angles $\varphi_{\text{I}} - \varphi_i$, the total angular momentum of the system,

$$L = n_1 + n_2 + \dots n_N + m, \qquad (A.4)$$

is an integral of motion, and we rewrite the basis functions as follows

$$e^{in_1\varphi_1 + in_2\varphi_2 + \dots + in_N\varphi_N}e^{im\varphi_{\text{I}}} = e^{in_1\varphi_1 + in_2\varphi_2 + \dots + in_N\varphi_N}e^{i(L - n_1 - n_2 \dots - n_N)\varphi_{\text{I}}}$$
$$= e^{in_1(\varphi_1 - \varphi_{\text{I}}) + in_2(\varphi_2 - \varphi_{\text{I}}) + \dots + in_N(\varphi_N - \varphi_{\text{I}})}e^{iL\varphi_{\text{I}}} = e^{in_1\tilde{\varphi}_1 + in_2\tilde{\varphi}_2 + \dots + in_N\tilde{\varphi}_N}e^{iL\varphi_{\text{I}}}, \qquad (A.5)$$

where $\tilde{\varphi}_i = \varphi_i - \varphi_{\text{I}}$. This allows us to work with a new set of coordinates $r_i/\xi = (|x_i|/\xi, \tilde{\varphi}_i)$, which should be accompanied by the transformation of the corresponding derivatives: $\frac{\partial}{\partial\varphi_i} = \frac{\partial}{\partial\tilde{\varphi}_i}$ and $\frac{\partial}{\partial\varphi_{\text{I}}} = -\sum_i^N \frac{\partial}{\partial\tilde{\varphi}_i} = -\sum_i^N \frac{\partial}{\partial\varphi_i}$.

The wave function of the system in the new coordinates has the form

$$\frac{1}{\xi^N}\tilde{\Psi}(\{x_i/\xi\},\varphi_I) = \frac{1}{\xi^N\sqrt{2\pi}}e^{i\varphi_I L}\tilde{\psi}(\{r_i/\xi\}) = \frac{1}{\sqrt{2\pi}}e^{i\varphi_I L}\prod_i^N\frac{1}{\xi}\tilde{f}(r_i/\xi), \qquad (A.6)$$

where in the last step, we introduced the mean-field treatment and the corresponding order parameter

$$f(r_i) = \frac{1}{\xi}\tilde{f}(r_i/\xi). \qquad (A.7)$$

All functions are normalized as $\|\Psi\| = 1$, $\|\Phi\| = 1$, and $\|f\| = 1$. To find the ground state of the Hamiltonian, we look for the minimum of the free energy

$$\begin{aligned}
F = & E - \mu N = \langle\Psi|\mathcal{H} - \mu N|\Psi\rangle = \left\langle\Psi\left|\frac{\hbar^2}{m_b\xi^2}\tilde{\mathcal{H}} - \mu N\right|\Psi\right\rangle \\
= & \left\langle\tilde{\Psi}\left|\frac{\hbar^2}{m_b\xi^2}\tilde{\mathcal{H}} - \mu N\right|\tilde{\Psi}\right\rangle = \frac{\hbar^2}{m_b\xi^2}\left\langle\tilde{\Psi}\left|\tilde{\mathcal{H}} - \frac{m_b\xi^2}{\hbar^2}\mu N\right|\tilde{\Psi}\right\rangle \equiv \frac{\hbar^2}{m_b\xi^2}\tilde{F},
\end{aligned} \qquad (A.8)$$

where $F$ is the free energy, $\tilde{F}$ is the dimensionless free energy and $\mu$ is the chemical potential. We calculate each component of the free energy as

$$\begin{aligned}
\frac{\hbar^2}{m_b\xi^2}\tilde{F}_1 \equiv & F_1 = \left\langle\Psi\left|-B_{\text{rot}}\frac{\partial^2}{\partial\varphi_I^2}\right|\Psi\right\rangle = \frac{1}{2\pi}\left\langle e^{i\varphi_I L}\prod_i^N f(r_i)\left|-B_{\text{rot}}\frac{\partial^2}{\partial\varphi_I^2}\right|e^{i\varphi_I L}\prod_i^N f(r_i)\right\rangle \\
= & -B_{\text{rot}}\Bigg[-2iL\sum_k\int dr_k f^*(r_k)\frac{\partial}{\partial\varphi_k}f(r_k) + \sum_k^N\int dr_k f^*(r_k)\frac{\partial^2}{\partial\varphi_k^2}f(r_k) \\
& -L^2 + \sum_k^N\sum_{l\neq k}^N\int\int dr_k dr_l f^*(r_k)f^*(r_l)\frac{\partial}{\partial\varphi_k}\frac{\partial}{\partial\varphi_l}f(r_k)f(r_l)\Bigg] \\
= & N\int dr f^*(r)\left[\frac{B_{\text{rot}}L^2}{N} + 2iB_{\text{rot}}L\frac{\partial}{\partial\varphi} - iB_{\text{rot}}A\frac{\partial}{\partial\varphi} - B_{\text{rot}}\frac{\partial^2}{\partial\varphi^2}\right]f(r), \qquad (A.9)
\end{aligned}$$

where

$$A = -i(N-1)\int dr f^*(r)\frac{\partial}{\partial\varphi}f(r), \qquad (A.10)$$

is the angular momentum of the bath, and

$$\frac{\hbar^2}{m_b\xi^2}\tilde{F}_2 \equiv F_2 = \left\langle\Psi\left|-\sum_i^N\frac{\hbar^2}{2m_b}\frac{\partial^2}{\partial r_i^2}\right|\Psi\right\rangle = N\int dr f^*(r)\left(-\frac{\hbar^2}{2m_b}\frac{\partial^2}{\partial r^2}\right)f(r), \qquad (A.11)$$

$$\begin{aligned}
\frac{\hbar^2}{m_b\xi^2}\tilde{F}_3 \equiv F_3 = & \left\langle\Psi\left|\sum_{i<j}^N g\delta(|r_i - r_j|)\right|\Psi\right\rangle \\
= & \sum_i\sum_{j<i}\int dr_i\int dr_j f^*(r_i)f^*(r_j)g\delta(|r_i - r_j|)f(r_i)f(r_i) \\
= & N\int f^*(r)\left(\frac{N-1}{2}g|f(r)|^2\right)f(r)dr, \qquad (A.12)
\end{aligned}$$

$$\frac{\hbar^2}{m_b\xi^2}\tilde{F}_4 \equiv F_4 = \left\langle\Psi\left|\sum_i U(r_i)\right|\Psi\right\rangle = N\int f^*(r)\big(U(r)\big)f(r)dr, \qquad (A.13)$$

where

$$U(r_i) = \alpha W(r_i, \tilde{\varphi}_i) + \frac{k|r_i|^2}{2}, \qquad \text{and} \qquad W(r_i, \tilde{\varphi}_i) = R(r_i) \cos(\tilde{\varphi}_i), \tag{A.14}$$

$$\frac{\hbar^2}{m_b \xi^2} \tilde{F}_5 \equiv F_5 = \langle \Psi | -\mu N | \Psi \rangle = N \int dr f^*(r)(-\mu) f(r), \tag{A.15}$$

and where $F$ is sum of the all $F_n$. Each of the free energy terms has the form

$$F_n = N \int dr f^*(r) \hat{O} f(r), \tag{A.16}$$

where $\hat{O}$ is a relevant operator.

To minimize the free energy, we vary $f^*$ and $f$ separately. Here, we consider the variation $f^* \to f^* + \delta f^*$

$$\delta F_n = F_n(f^* + \delta f^*) - F_n(\delta f^*) = cN \int dr \delta f^*(r) \hat{O} f(r), \tag{A.17}$$

where $c$ equals 1 if $\hat{O}$ is independent of $f$ and 2 otherwise, i.e., if the term is non-linear in $f$. The variation of $F$ with respect to $f^*$ fixed to zero is equivalent for the same condition for $\tilde{F}$ and $\tilde{f}^*$ that results in

$$\begin{aligned} 0 = \quad & N \int \frac{dr}{\xi^2} \delta \tilde{f}^*(r/\xi) \Big[ iL2B_{\text{rot}} \frac{m_b \xi^2}{\hbar^2} \frac{\partial}{\partial \varphi} - i2B \frac{m_b \xi^2}{\hbar^2} A \frac{\partial}{\partial \varphi} - B_{\text{rot}} \frac{m_b \xi^2}{\hbar^2} \frac{\partial^2}{\partial \varphi^2} \\ & + \frac{m_b \xi^2}{\hbar} \frac{B_{\text{rot}} L^2}{N} - \frac{1}{2} \frac{\partial^2}{\partial (r/\xi)^2} + (N-1) g \frac{m_b}{\hbar^2} \big| \tilde{f}(r/\xi) \big|^2 \\ & + \frac{m_b \xi^4}{\hbar^2} \frac{k|r/\xi|^2}{2} + \frac{m_b \xi^2}{\hbar^2} \alpha W(r/\xi) - \frac{m_b \xi^2}{\hbar^2} \mu \Big] \tilde{f}(r/\xi). \end{aligned} \tag{A.18}$$

From this expression, we obtain the Gross-Pitaevskii-like equation

$$\begin{aligned} \Big[ & i2B_{\text{rot}} \frac{m_b \xi^2}{\hbar^2} (L - A) \frac{\partial}{\partial \varphi} - B_{\text{rot}} \frac{m_b \xi^2}{\hbar^2} \frac{\partial^2}{\partial \varphi^2} + \frac{m_b \xi^2}{\hbar^2} \frac{B_{\text{rot}} L^2}{N} \\ & - \frac{1}{2} \frac{\partial^2}{\partial (r/\xi)^2} + (N-1) \frac{m_b}{\hbar^2} g \big| \tilde{f}(r/\xi) \big|^2 \\ & + \frac{m_b \xi^4}{\hbar^2} \frac{k|r/\xi|^2}{2} + \frac{m_b \xi^2}{\hbar^2} \alpha W(r/\xi) \Big] \tilde{f}(r/\xi) = \frac{m_b \xi^2}{\hbar^2} \mu \tilde{f}(r/\xi), \end{aligned} \tag{A.19}$$

which leads to Eq. (6).

The solution to the GPE allows us to calculate all relevant observables. For example, the energy is given by the expectation value of the Hamiltonian from Eq. (A.1)

$$E = \langle \Psi | \mathcal{H} | \Psi \rangle \equiv \frac{\hbar^2}{m_b \xi^2} \langle \tilde{\Psi} | \tilde{\mathcal{H}} | \tilde{\Psi} \rangle. \tag{A.20}$$

To calculate it, we use Eqs. (A.9)-(A.15)

$$\begin{aligned} E = \quad & -2B_{\text{rot}} LA + B_{\text{rot}} A^2 - B_{\text{rot}} N \int f^*(r) \frac{\partial^2}{\partial \varphi^2} f(r) dr - N \frac{1}{2} \int f^*(r) \frac{\partial^2}{\partial r^2} f(r) dr \\ & + B_{\text{rot}} L^2 + \frac{N(N-1)}{2} g \int |f(r)|^4 dr + N \int |f(r)|^2 U(r) dr. \end{aligned} \tag{A.21}$$

We compare this equation to the integrated Eq. (A.18) to derive the expression for the energy

$$E = N\mu - B_{\text{rot}} A^2 - \frac{1}{2} g N(N-1) \int |f(r)|^4 dr, \tag{A.22}$$

which is used to calculate the rotational spectrum

$$\Delta E_L = E(L) - E(L=0). \tag{A.23}$$

We expect that for $L \to 0$

$$\Delta E_L = B_{\text{eff}} L^2, \qquad \frac{\partial E}{\partial L} = 2 B_{\text{eff}} L. \tag{A.24}$$

To connect the effective rotational constant $B_{\text{eff}}$ to the angular momentum of the impurity, we calculate the derivative over $L$

$$\frac{\partial}{\partial L} \mathcal{H}_L = \frac{\partial}{\partial L} \left( e^{-i \varphi_I L} \mathcal{H} e^{i \varphi_I L} \right) = \frac{\partial}{\partial L} \left( e^{-i \varphi_I L} \left( -B_{\text{rot}} \frac{\partial^2}{\partial \varphi_I^2} \right) e^{i \varphi_I L} \right), \tag{A.25}$$

where

$$\frac{\partial^2}{\partial \varphi_I^2} e^{i \varphi_I L} = e^{i \varphi_I L} \left( \frac{\partial^2}{\partial \varphi_I^2} + 2 i L \frac{\partial}{\partial \varphi_I} - L^2 \right). \tag{A.26}$$

We conclude that

$$\frac{\partial}{\partial L} \mathcal{H}_L = 2 B_{\text{rot}} \left( L - 2 i \frac{\partial}{\partial \varphi_I} \right). \tag{A.27}$$

According to the Hellmann-Feynman theorem [42],

$$\frac{\partial E}{\partial L} = \left\langle \Psi \left| \frac{\partial}{\partial L} \mathcal{H}_L \right| \Psi \right\rangle = 2 B_{\text{rot}} \left( L - i \left\langle \psi \left| -\sum_i^N \frac{\partial}{\partial \varphi_i} \right| \psi \right\rangle \right) =$$

$$= 2 B_{\text{rot}} \left( L + i N \left\langle f \left| \frac{\partial}{\partial \varphi} \right| f \right\rangle \right) = 2 B_{\text{rot}} (L - A) = 2 B_{\text{rot}} J, \tag{A.28}$$

which leads to Eq. (9) of the main text $B_{\text{eff}} = B_{\text{rot}} J / L$ and $\Delta E_L = B_{\text{rot}} J L$.

# B  Numerical method and its convergence

We find the lowest-energy solution of Eq. (6) numerically using the imaginary time evolution. Namely, we use the finite difference method to discretize space coordinates and approximate the spacial differential operators. We reduce a two-dimensional problem to a one-dimensional one by the natural ordering of the grid points by the transformation of each point from $(x_i, y_j)$ to $z_{N_{\text{grid}} i + j}$, where $N_{\text{grid}}$ is a grid dimension in a single direction. We discretize spacial derivatives using central finite difference.

In the next step, we focus on the imaginary time derivative. Equation (6) can be expressed in general form

$$\left( \hat{D}_r + \mathcal{F}[f(r,t)] + V(r) \right) f(r,t) = i \frac{\partial}{\partial t} f(r,t), \tag{B.1}$$

where $D_r$ is an operator representing all of spacial derivatives, $\mathcal{F}$ is a functional representation of all non-linear terms, and $V$ is a potential term. Employing the semi-implicit backward Euler scheme (which proved efficient in the non-linear equations' imaginary time evolution, especially rotating GPEs [43]), we discretize the time derivative and linearize the equation by changing the non-linear term argument to the order parameter from next time step

$$\left( \hat{D}_r + \mathcal{F}\left[ f(r,t)^{n+1} \right] + V(r) \right) f(r,t)^n = \frac{i}{\Delta t} f(r,t)^{n+1} - \frac{i}{\Delta t} f(r,t)^n, \tag{B.2}$$

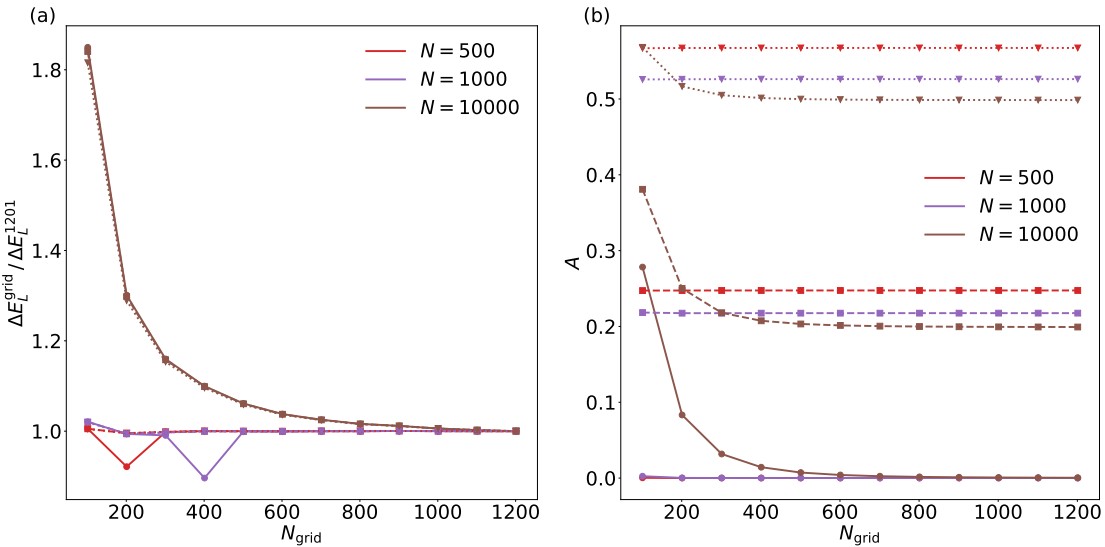

Figure 8: (a) Rotational energy $E_L$ and (b) the angular momentum of the bath $A$ convergences with the grid size for different number of bosons $N$ and interaction strengths $\alpha = 0.1\, m_b\xi^2/\hbar^2$ (solid line with rounds markers), $\alpha = 5\, m_b\xi^2/\hbar^2$ (dashed line with square markers), $\alpha = 10\, m_b\xi^2/\hbar^2$ (dotted line with triangle markers).

where the upper index $n$ denotes the time step. Then

$$\left(\frac{i\mathbb{1}}{\Delta t} + \hat{D}_r + \mathcal{F}\big[f(r,t)^{n+1}\big] + V(r)\right)f(r,t)^n = \frac{i}{\Delta t}f(r,t)^{n+1}. \tag{B.3}$$

The scheme is time-symmetric, and we can reverse $t \to -t$ and $n+1 \to n$. As a result, we get

$$\left(\frac{i\mathbb{1}}{\Delta t} + \hat{D}_r + \mathcal{F}\big[f(r,t)^n\big] + V(r)\right)f(r,t)^{n+1} = \frac{i}{\Delta t}f(r,t)^n. \tag{B.4}$$

If we change a real time to an imaginary time $\Delta t = i\Delta\tau$, we obtain

$$\left(\frac{\mathbb{1}}{d\tau} + \hat{D}_r + \mathcal{F}\big[f(r,t)^n\big] + V(r)\right)f(r,t)^{n+1} = \frac{1}{d\tau}f(r,t)^n. \tag{B.5}$$

Due to the spatial discretization, the above problem can be formulated in linear matrix equation form

$$\mathbb{A}x = b, \tag{B.6}$$

where $\mathbb{A}$ is a sparse matrix and $x$, $b$ are vectors. Because of the sparsity of the $\mathbb{A}$ matrix, we can employ robust Krylov solvers to solve the equation [44]. Those powerful interactive methods solve linear and eigenvalue problems in memory and time-efficient ways. In our work, we use the Krylov solver implemented in KrylovKit.jl [45]. The code in Julia programming language that can allow one to reproduce our results is available on GitLab [27]. Note that our open-source program can be extended to solve other non-linear GP-type equations for different dimensionality.

In Figure 8, we show the convergence of the rotational energy $\Delta E_L$ and bath's angular momentum $A$ for different condensate sizes and interaction strengths. We see that while the strength of the interaction affects the convergence only weakly, the increasing size of the condensate requires a bigger computational grid to obtain the exact energy of the system. In our calculations, we use $N_{\mathrm{grid}} = 801$, which might overestimate the energy for bigger condensate

sizes. With the increasing size of the condensate, the region when the deformation of the order parameter is present becomes smaller with respect to the whole condensate and, in effect, to the computational grid. Therefore, the numerical calculation of the energy and angular momentum becomes less accurate. We choose $N_{\text{grid}} = 801$ as an optimal grid size considering that computation time and memory requirements that scale approximately as $N_{\text{grid}}^2$. According to Fig. 8, this choice allows us to calculate the properties of the angulon regime with $N \sim 1000$ accurately.

## C  Two-body problem

### C.1  Weakly-interacting limit

Here, we consider the two-body problem of a single boson interacting with a molecule:

$$\left[B_{\text{rot}}L^2 + 2iB_{\text{rot}}L\frac{\partial}{\partial\varphi} - B_{\text{rot}}\frac{\partial^2}{\partial\varphi^2} - \frac{\hbar^2}{2m_b}\frac{\partial^2}{\partial r^2} + \alpha W(r)\right]f(r) = Ef(r),\qquad\text{(C.1)}$$

in the limit $\alpha \to 0$. We follow the approach proposed to solve such a problem in Ref. [29]. The wave function can be expanded as

$$f(\tilde{r}) = \sqrt{\frac{r_0}{x}}\sum_{m=-1}^{m=1}a_m f_m(\tilde{x})e^{im\varphi},\qquad\text{(C.2)}$$

where $\tilde{r} = (\tilde{x}, \varphi)$ is a reduced relative coordinate, and $\tilde{x} = x/r_0$. We limit the expansion to $|m| = 0, 1$ because of the form of considered potential. Then, the system is described by the equation

$$\frac{\partial^2}{\partial\tilde{x}^2}f_m(\tilde{x}) + \frac{1-4m^2}{4\tilde{x}^2}f_m(\tilde{x}) + \epsilon_{m-L}^2 f_m = \tilde{\alpha}\sum_l\frac{a_l}{a_m}f_l(\tilde{x})V_{ml}(\tilde{x}),\qquad\text{(C.3)}$$

where $\epsilon_k^2 = \frac{m_b}{\hbar^2 r_0^2}(2E - 2B_{\text{rot}}k^2)$, $\tilde{\alpha} = \frac{m_b}{\hbar^2 r_0^2}\alpha$, and

$$V_{ml}(\tilde{x}) = \frac{1}{\pi}\int_0^{2\pi}e^{i(l-m)\varphi}W(\tilde{x}, \varphi)\mathrm{d}\varphi,\qquad\text{(C.4)}$$

is a potential matrix element. The solution to this equation is

$$f_m(\epsilon_{m-L}, \tilde{x}) = F_m(\epsilon_{m-L}, \tilde{x}) - \tilde{\alpha}\sum_l\frac{a_l}{a_m}\int_0^{\tilde{x}}g_m(\epsilon_{m-L}, \tilde{x}, \tilde{x}')V_{ml}(\tilde{x}')f_l(\epsilon_{l-L}, \tilde{x}')\mathrm{d}\tilde{x}',\qquad\text{(C.5)}$$

where $F_m(\epsilon_{m-L}, \tilde{x}) = \sqrt{\tilde{x}}J_{|m|}(\epsilon_{m-L}\tilde{x})(2/\epsilon_{m-L})^{|m|}|m|!$ is the solution of the free Schrodinger equation in terms of Bessel function $J_m$, and

$$g_m(\epsilon_{m-L}, \tilde{x}, \tilde{x}') = \frac{i\pi}{4}\sqrt{\tilde{x}\tilde{x}'}\left[H_{|m|}^{(1)}(\epsilon_{m-L}\tilde{x})H_{|m|}^{(2)}(\epsilon_{m-L}\tilde{x}') - H_{|m|}^{(1)}(\epsilon_{m-L}\tilde{x}')H_{|m|}^{(2)}(\epsilon_{m-L}\tilde{x})\right],\quad\text{(C.6)}$$

is a Green's function where $H_m^{(n)}$ are Hankel functions. From the boundary conditions of the free wave function and Green's function, the condition for the bound state for our potential is given as

$$\begin{aligned}c_{1,0}a_0 + a_1 &= 0,\\ c_{0,1}a_1 + a_0 + c_{0,-1}a_{-1} &= 0,\\ c_{-1,0}a_0 + a_{-1} &= 0,\end{aligned}\qquad\text{(C.7)}$$

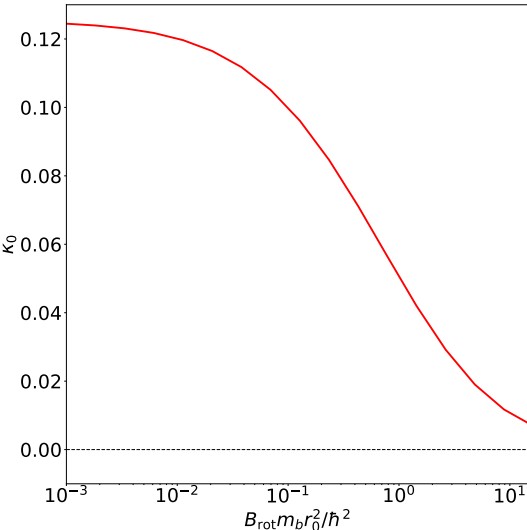

Figure 9: Parameter $\kappa_0$ as a function of the rotational constant $B_{\text{rot}}$ for $L = 0$.

which gives a solution

$$c_{0,1}c_{1,0} + c_{0,-1}c_{-1,0} = 1, \tag{C.8}$$

where

$$c_{ml} = \tilde{\alpha}\left(\frac{\epsilon_{m-L}}{2}\right)^{|m|} \frac{i\pi}{2|m|!} \int_0^\infty \sqrt{\tilde{x}'} H_{|m|}^{(1)}(\epsilon_{m-L}\tilde{x}') V_{ml}(\tilde{x}') f_l(\epsilon_{l-L}, \tilde{x}') \mathrm{d}\tilde{x}', \tag{C.9}$$

where $H_m^{(n)}$ is the Hankel function. We assume limits $\tilde{\alpha} \to 0$ and $E \to 0$. Note, that then $\epsilon_0 \to 0$ and $\epsilon_k^2 \to -2\frac{m_b}{\hbar^2 r_0^2} B_{\text{rot}} k^2$. Now, we separately consider two cases of $L = 0$ and $L = 1$. In the former case, Eq. (C.8) can be rewritten in form

$$\tilde{\alpha}^2 \kappa_0 \ln \epsilon_0 \simeq -1, \tag{C.10}$$

which leads to $\epsilon_0 \propto e^{-\frac{1}{\tilde{\alpha}^2 \kappa_0}}$, where

$$\kappa_0 = \frac{i\pi}{4}\left(\epsilon_1 \kappa_{0,1} + \epsilon_{-1} \kappa_{0,-1}\right), \tag{C.11}$$

with

$$\kappa_{0,m} = \int_0^\infty \sqrt{\tilde{x}'} H_{|m|}^{(1)}(\epsilon_m \tilde{x}') V_{m0}(\tilde{x}') f_0(\epsilon_0 \tilde{x}') \mathrm{d}\tilde{x}' \int_0^\infty \sqrt{\tilde{x}'} V_{0m}(\tilde{x}') f_m(\epsilon_m \tilde{x}') \mathrm{d}\tilde{x}'. \tag{C.12}$$

The system is bound when the value of $\kappa_0 > 0$. In Figure 9, we show the value of $\kappa_0$ as a function of the rotational constant $B_{\text{rot}}$. For $L = 0$, the system is always bound.
In second case ($L = 1$), we obtain condition

$$\epsilon_0 \ln \epsilon_0 \simeq \frac{\tilde{\alpha}^2 \gamma - 1}{\tilde{\alpha}^2 \kappa_1}, \tag{C.13}$$

where $\tilde{\alpha}^2 \gamma = c_{0,-1}c_{-1,0}$. Both parameters $\gamma$ and $\kappa_1$ have finite values, and with $\tilde{\alpha} \to 0$, the above condition is never fulfilled. From that, we conclude that the bound state does not exist in the limit $\alpha \to 0$.

## C.2 Strongly-interacting limit

To study Eq. (C.1) in the limit $\alpha \to \infty$, we approximate the Hamiltonian assuming that the boson moves only around the minimum of the impurity-boson potential, i.e., $r = r_{\min}$. This assumption allows us to re-write Eq. (C.1) as

$$\left[ B_{\rm rot} L^2 + 2i B_{\rm rot} L \frac{\partial}{\partial \varphi} - B \frac{\partial^2}{\partial \varphi^2} - \frac{\hbar^2}{2m_b} \frac{\partial^2}{\partial r^2} - \frac{\hbar^2}{2m_b r} \frac{\partial}{\partial r} + \alpha W(r) \right] f(r) = E f(r), \quad \text{(C.14)}$$

where $B = B_{\rm rot} + \frac{\hbar^2}{2m_b r_{\min}^2}$. As a next step, we use a transformation $f = e^{iB_{\rm rot}\varphi/B} \tilde{f}$, where $\tilde{f}$ solves the equation

$$\left[ -B \frac{\partial^2}{\partial \varphi^2} - \frac{\hbar^2}{2m_b} \frac{\partial^2}{\partial r^2} - \frac{\hbar^2}{2m_b r} \frac{\partial}{\partial r} + \alpha W(r) \right] \tilde{f}(r) = \tilde{E} \tilde{f}(r), \quad \text{(C.15)}$$

with $\tilde{E} = E - B_{\rm rot} L^2 + \frac{(B_{\rm rot} L)^2}{B}$. The impurity is allowed to move only close to the position of the minimum in the angular space. In other words, $\tilde{f}$ vanishes at $\varphi = 0$ independent of $L$. As Eq. (C.15) and the corresponding boundary conditions are independent of $L$, we conclude that $\tilde{f}$ does not depend on $L$. The energy $\Delta E_L$ can now be written as

$$\Delta E_L = \left( B_{\rm rot} - \frac{B_{\rm rot}^2}{B} \right) L^2, \quad \text{(C.16)}$$

where the quantity in the parenthesis defines the effective rotational constant.

# D Angulon in thermodynamic limit

To analyze Eq. (A.20) in the thermodynamic limit, we consider the limit $k \to 0$

$$\left[ i2B_{\rm rot} J \frac{\partial}{\partial \varphi} - B_{\rm rot} \frac{\partial^2}{\partial \varphi^2} - \frac{\hbar^2}{2m_b} \frac{\partial^2}{\partial r^2} + (N-1)g|f(r)|^2 + \alpha R(r) \cos \varphi \right] f(r) = \tilde{\mu} f(r), \quad \text{(D.1)}$$

where $R(r) = \frac{r}{r_0} e^{-r^2/r_0^2}$ and $\tilde{\mu} = \mu - \frac{B_{\rm rot} L^2}{N}$. We look for a solution in the form

$$f(r) = f_0(r) + f_{\rm r}(r) + i f_{\rm i}(r), \quad \text{(D.2)}$$

where $f_0(r) = \sqrt{\frac{\mu}{Ng}}$ is the Thomas-Fermi profile in the limit $k \to 0$. The real and imaginary parts of $f$ obey the coupled system of equations

$$-2B_{\rm rot} J \frac{\partial}{\partial \varphi} f_{\rm i} - B_{\rm rot} \frac{\partial^2}{\partial \varphi^2} f_{\rm r} - \frac{\hbar^2}{2m_b} \frac{\partial^2}{\partial r^2} f_{\rm r} + g(N-1)(f_0^3 + 3f_0^2 f_{\rm r} + 3f_0 f_{\rm r}^2 + f_{\rm r}^3)$$
$$+ (N-1)g f_{\rm i}^2 f_0 + (N-1)g f_{\rm i}^2 f_{\rm r} + \alpha R(r) \cos \varphi f_0 + \alpha R(r) \cos \varphi f_{\rm r} = \mu f_0 + \mu f_{\rm r}, \quad \text{(D.3)}$$

and

$$2B_{\rm rot} J \frac{\partial}{\partial \varphi} f_{\rm r} - B_{\rm rot} \frac{\partial^2}{\partial \varphi^2} f_{\rm i} - \frac{\hbar^2}{2m_b} \frac{\partial^2}{\partial r^2} f_{\rm i} + g(N-1) f_{\rm i}^3$$
$$+ (N-1)g(f_0^2 + 2f_0 f_{\rm r} + f_{\rm r}^2) f_{\rm i} + \alpha R(r) \cos \varphi f_{\rm i} = \mu f_{\rm i}. \quad \text{(D.4)}$$

As we focus on the limit $\alpha \to 0$, we assume that $f_{\rm r} \propto \alpha$ and $f_{\rm i} \propto \alpha$, and get rid of the terms of higher orders in $\alpha$. As a result, we derive the two equations

$$-\frac{\hbar^2}{2m_b} \frac{\partial^2}{\partial r^2} f_{\rm r} - B_{\rm rot} \frac{\partial^2}{\partial \varphi^2} f_{\rm r} + 2\mu f_{\rm r} = 2B_{\rm rot} J \frac{\partial}{\partial \varphi} f_{\rm i} - \alpha R(r) \cos \varphi f_0, \quad \text{(D.5)}$$

and

$$-\frac{\hbar^2}{2m_b}\frac{\partial^2}{\partial r^2}f_i - B_{\mathrm{rot}}\frac{\partial^2}{\partial \varphi^2}f_i = -2B_{\mathrm{rot}}J\frac{\partial}{\partial \varphi}f_r. \tag{D.6}$$

The solutions to these equations can be written as

$$f_r = c_r(r)\cos\varphi \qquad \text{and} \qquad f_i = c_i(r)\sin\varphi, \tag{D.7}$$

where the functions $c_r$ and $c_i$ obey the equations

$$-\frac{\hbar^2}{2m_b}\frac{\partial^2}{\partial r^2}c_r(r) - \frac{\hbar^2}{2m_b r}\frac{\partial}{\partial r}c_r(r) + \left(B_{\mathrm{rot}} + 2\mu + \frac{\hbar^2}{2m_b r^2}\right)c_r(r) = 2B_{\mathrm{rot}}Jc_i(r) - \alpha R(r)f_0, \tag{D.8}$$

and

$$-\frac{\hbar^2}{2m_b}\frac{\partial^2}{\partial r^2}c_i(r) - \frac{\hbar^2}{2m_b r}\frac{\partial}{\partial r}c_i(r) + \left(B_{\mathrm{rot}} + \frac{\hbar^2}{2m_b r^2}\right)c_i(r) = 2B_{\mathrm{rot}}Jc_r(r). \tag{D.9}$$

To find the solution, we decouple the equations assuming that $f_r$ does not depend on $J$. After a few transformations, we derive

$$r^2\frac{\partial^2}{\partial r^2}c_r(r) + r\frac{\partial}{\partial r}c_r(r) - \left[\frac{m_b}{\hbar^2}(2B_{\mathrm{rot}} + 4\mu)r^2 + 1\right]c_r(r) = 2\frac{m_b}{\hbar^2}\alpha R(r)f_0 r^2, \tag{D.10}$$

and

$$r^2\frac{\partial^2}{\partial r^2}c_i(r) + r\frac{\partial}{\partial r}c_i(r) - \left[2\frac{m_b}{\hbar^2}B_{\mathrm{rot}}r^2 + 1\right]c_i(r) = -4\frac{m_b}{\hbar^2}B_{\mathrm{rot}}Jc_r(r)r^2, \tag{D.11}$$

which are the modified Bessel equations. The general form of the equation is

$$r^2\frac{\partial^2}{\partial r^2}h(r) + r\frac{\partial}{\partial r}h(r) - \left(a^2 r^2 + 1\right)h(r) = g(r). \tag{D.12}$$

The general solution is given as

$$h(r) = c_1 I_1(ar) + c_2 K_1(ar), \tag{D.13}$$

where $I_1(r)$ and $K_1(r)$ are the modified Bessel functions of first and second kind. Associated Green's function is

$$G(r,r') = \begin{cases} c(r')I_1(ar)K_1(ar'), & \text{if } 0 < r < r' < \infty \\ c(r')K_1(ar)I_1(ar'), & \text{if } 0 < r' < r < \infty \end{cases} \tag{D.14}$$

From

$$\lim_{r\to r'^+}G(r,r') - \lim_{r\to r'^-}G(r,r') = \frac{1}{r'^2}, \tag{D.15}$$

with Wronskian

$$\mathcal{W}(r') = \frac{1}{r'}, \tag{D.16}$$

we derive

$$c(r') = \frac{1}{r'}. \tag{D.17}$$

After incorporating the proper limits of the solution, we arrive at

$$c_r(r) = -2\frac{m_b}{\hbar^2}\alpha\sqrt{\frac{\mu}{Ng}}F_r(r), \tag{D.18}$$

$$c_i(r) = -8\frac{m_b^2}{\hbar^4}B_{\mathrm{rot}}J\alpha\sqrt{\frac{\mu}{Ng}}F_i(r), \tag{D.19}$$

where

$$F_{\rm r}(r) = I_1(br) \int_r^\infty K_1(br')R(r')r'{\rm d}r' + K_1(br) \int_0^r I_1(br')R(r')r'{\rm d}r', \tag{D.20}$$

and

$$F_{\rm i}(r) = I_1(cr) \int_r^\infty K_1(cr')F_{\rm r}(r')r'{\rm d}r' + K_1(cr) \int_0^r I_1(cr')F_{\rm r}(r')r'{\rm d}r', \tag{D.21}$$

and $b = \sqrt{\frac{m_b}{\hbar^2} 2B_{\rm rot}}$, $c = \sqrt{\frac{m_b}{\hbar^2}(2B_{\rm rot} + 4\mu)}$. Finally, we have a full solution

$$f_{\rm r}(\boldsymbol{r}) = -2\frac{m_b}{\hbar^2}\alpha\sqrt{\frac{\mu}{Ng}}F_{\rm r}(r)\cos\varphi, \tag{D.22}$$

$$f_{\rm i}(\boldsymbol{r}) = -8\frac{m_b^2}{\hbar^4}B_{\rm rot}J\alpha\sqrt{\frac{\mu}{Ng}}F_{\rm i}(r)\sin\varphi, \tag{D.23}$$

Now, we can calculate the angular momentum of the bath

$$A = -i(N-1)\int {\rm d}\boldsymbol{r}f^*\frac{\partial}{\partial\varphi}f = (N-1)\int {\rm d}\boldsymbol{r}(f_{\rm r} - if_{\rm i})\left(if_{\rm r}\frac{\sin\varphi}{\cos\varphi} + f_{\rm i}\frac{\cos\varphi}{\sin\varphi}\right) =$$

$$= (N-1)\int {\rm d}\boldsymbol{r}\left(f_{\rm r}f_{\rm i}\frac{\sin\varphi}{\cos\varphi} + f_{\rm r}f_{\rm i}\frac{\cos\varphi}{\sin\varphi}\right) = (N-1)\int {\rm d}\boldsymbol{r}f_{\rm r}f_{\rm i}\left(\frac{\sin^2\varphi + \cos^2\varphi}{\cos\varphi\sin\varphi}\right) =$$

$$= 32\pi\frac{m_b^3}{\hbar^6}\frac{\mu}{g}B_{\rm rot}J\alpha^2\int_0^\infty {\rm d}rF_{\rm r}(r)F_{\rm i}(r)r = 32\pi\frac{m_b^3}{\hbar^6}\frac{\mu}{g}B_{\rm rot}J\alpha^2 H_{\rm ri}. \tag{D.24}$$

According to the definition $J = L - A$ so we have

$$A = \frac{32\pi\frac{m_b^3}{\hbar^6}\frac{\mu}{g}B_{\rm rot}\alpha^2 H_{\rm ri}}{1 + 32\frac{m_b^3}{\hbar^6}\pi\frac{\mu}{g}B_{\rm rot}\alpha^2 H_{\rm ri}}L, \tag{D.25}$$

and

$$J = \frac{1}{1 + 32\pi\frac{m_b^3}{\hbar^6}\frac{\mu}{g}B_{\rm rot}\alpha^2 H_{\rm ri}}L. \tag{D.26}$$

We use a formula derived from the Hellmann-Feynman theorem

$$B_{\rm eff}L^2 = B_{\rm rot}JL, \tag{D.27}$$

to find a rotational constant for $L \to 0$

$$B_{\rm eff} = \frac{1}{1 + 32\pi\frac{m_b^3}{\hbar^6}\frac{\mu}{g}B_{\rm rot}\alpha^2 H_{\rm ri}}B_{\rm rot}. \tag{D.28}$$

If, instead, we use the explicit formula for rotational energy

$$\Delta E_L = B_{\rm rot}L^2 - B_{\rm rot}A^2 - \frac{1}{2}gN(N-1)\left(\int |f|^4 - \int |f_0 + f_{\rm r}|^4\right), \tag{D.29}$$

we would not obtain the correct result. To understand that, we need to remember that in our previous approximation, we assumed that $f_{\rm r}$ does not depend on $J$. If we look for that correction and look at the order parameter in the form

$$f = f_0 + f_{\rm r}(\alpha) + \delta f_{\rm r}(\alpha, L) + if_{\rm i}(L), \tag{D.30}$$

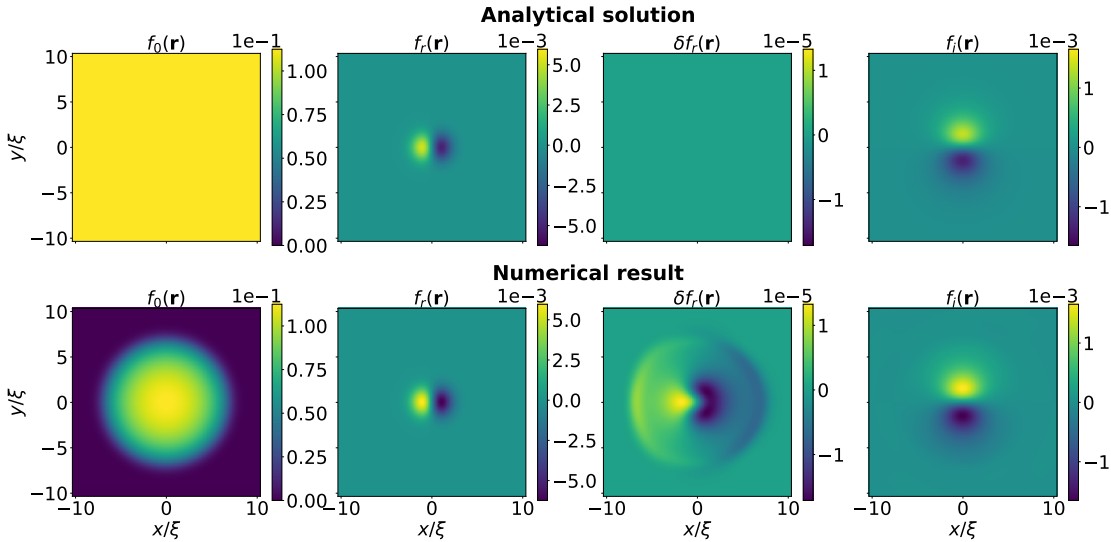

Figure 10: Top row: components of the order parameter for $\alpha = 2\sqrt{2}$, $L = 1$. Lower row: the numerical results for $n = n_0$, where $f_0 = f(\alpha = 0, L = 0)$, $f_{\rm r} = {\rm Re}\big[f(\alpha = 2\sqrt{2}, L = 0) - f_0\big]$, $\delta f_{\rm r} = {\rm Re}\big[f(\alpha = 2\sqrt{2}, L = 1) - f_{\rm r}\big]$, and $f_{\rm i} = {\rm Im}\big[f(\alpha = 2\sqrt{2}, L = 1)\big]$. Note different scales in each column.

we see that

$$\int |f|^4 - \int |f_0 + f_{\rm r}|^4 \simeq \int 2f_{\rm i}^2 f_0^2 + \int 4\delta f_{\rm r} f_0^3, \tag{D.31}$$

where both terms are of the same magnitude and opposite sign. Our solution is missing the second term, leading to the wrong estimation of the rotational energy. However, this term does not appear in calculations of the angular momentum, and Eq. (D.28) is not affected by this problem.

In Figure 10, we compare the order parameter between numerical results and analytical solution. It shows that our analytical solution does not include $\delta f_{\rm r}(r)$ due to angular momentum $L$, which is two orders of the magnitude smaller than the deformation of the real part due to interaction $f_{\rm r}(r)$ and of the imaginary part due to interaction and angular momentum $f_{\rm i}(r)$.

# E   Increasing the angular momentum in the few-body regime

We show the rotational energy and molecule angular momentum in the few-body regime in Fig. 11. The results show no critical velocity in the system as angulon is not formed. The properties of the system are driven by the harmonic trap.

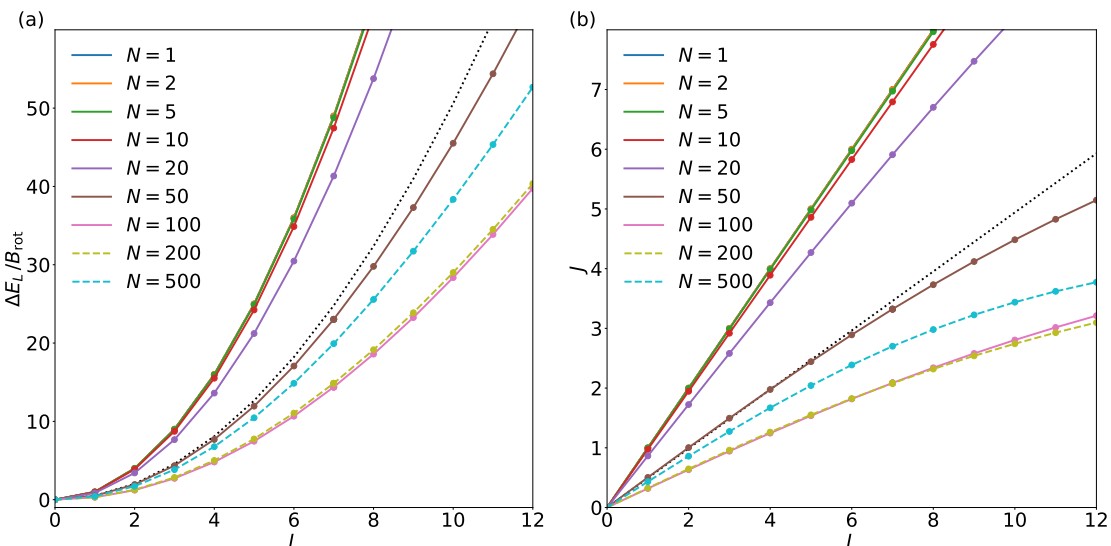

Figure 11: (a) The rotational energy $\Delta E_L$ and (b) the angular momentum of the molecule a few-body regime. The gray dashed line shows the free rotor results, and the black dotted line shows analytical results for the angulon regime.

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
