# Peer review of "Quantum rotor in a two-dimensional mesoscopic Bose gas"

_SciPost Physics, doi:SciPost Phys. 18, 059 (2025)_

## Round 1 · Referee Report · Anonymous (Referee 1) · 2024-8-20

Strengths

  1. clear problem statement
  2. simple and powerful method explored
  3. clear physical interpretation of the obtained results
  4. plenty of new interesting results obtained

Weaknesses

I have some tiny remarks (please see the pdf file) but I would have not called them true weaknesses.

Report

I think the journal acceptance criteria are fully met.

Requested changes

  1. The physical origin of the impurity-boson interaction is unclear. The manuscript suggests the rotor model for the impurity. Are bosons also rotors?

  2. On p.~4 we read: ``The normalized function $\psi$ defines the probability of finding a boson at a given position in a molecular frame of reference. It does not depend explicitly on the angle $\varphi_I$...'', however, coordinates ${\bf r}_i$ explicitly suggest such dependence. I think this issue should be clarified in the revised manuscript.

  3. What do the authors mean by the following sentence: ``Indeed, a strong deformation of the density of the Bose gas requires a number of phonons in momentum space for its description.''?

  4. The breakdown of the angulon behavior at large angular momenta (see Fig.~6) is signaled as a deviation of the rotation energy from the quadratic law. However, at large $L$ the energy $\Delta E^{num}_L$ seems to support a linear dependence. Is it really linear, and is there some physical reasoning for it?

Recommendation

Publish (easily meets expectations and criteria for this Journal; among top 50%)

  • validity: top
  • significance: high
  • originality: top
  • clarity: top
  • formatting: excellent
  • grammar: -

Author:  Michał Suchorowski  on 2024-11-26  [id 4991]

(in reply to Report 1 on 2024-08-20)

Dear Referee,

We thank you for your comments. In addition to the comments provided with the resubmitted manuscript, we address them in more detail in the attached file. We hope that the revised manuscript is ready for publication.

Sincerely,
The Authors

Attachment:

Reply__Report_1.pdf

---

## Round 1 · Referee Report · Anonymous (Referee 2) · 2024-9-30

Strengths

1-clear, 2-well organized, 3- a good combination of analytical results and numerics.

Weaknesses

1 - it is not clear what aspects of the problem depend on the number of particles in the system and what on the harmonic confinement.

Report

This paper is well-organized and clearly written. While some of the results are quite technical, the authors explain them in simple terms. The description of the angulon quasiparticle in real space in terms of the Gross-Pitaevskii equation (GPe) is new and potentially very useful, given that it allows one to capture the physics of non-uniform condensates in the presence of the strong interactions between the boson and the impurity.

The other strength of the manuscript is that the authors consider the system in a harmonic trap so that it can be realized in the experiment, however, this strength also turns out to be its major weakness, because the effects of the trap on the angulon physics are somewhat obscure. In particular, the authors consider the regime where the central density in the trap remains fixed, so for a small number of particles in the system, the potential has to be more confining, and the effects of the trap are important. At the same time, the effects of the trap should be less important as $N \to \infty$. The main analytical result is Eq.(18) and it is derived under the assumption that $N$ is finite, but the trap is turned off. This analytical result agrees well with numerics for a range of $N\gg1$, but then suddenly breaks down when $N$ becomes larger than some $N^*$. The fact that such behavior does not seem to follow from the formalism used (from what I understand the result in Eq.(18) should hold for arbitrary $N\gg1$), makes the whole approach look questionable. Since it is not clear what triggers such a behavior, it is not clear whether the harmonic trap has to do with it.

Just to sum up the above, when the authors talk about different regimes based on the number of particles in the system, it is hard to tell whether this is something inherent to the angulon problem, or some of the regimes are the artifacts of the harmonic confinement.

I think to make the discussion clearer and make the claims of the authors stronger, they should also consider the problem in the absence of harmonic confinement, for example, by putting the system into a box of finite size. Then one can fix the density and study the problem as a function of $N$ and the box size $L$. If the results in this system agree with the ones in the current manuscript in the regime where $N\gg1$, then I will accept the validity of the presented results and will be happy to recommend this manuscript to be published in SciPost.

Finally, the angulon problem is a close cousin of the Bose polaron problem, as was also pointed out by the authors. For the Bose polarons, there is a thermodynamic relation between the number of particles inside the polaronic cloud and its energy, see https://doi.org/10.1103/PhysRevLett.126.123403. Can one expect something similar to hold for angulons?

Requested changes

1-add results/discussion of the angulon physics in the absence of the harmonic confinement and compare with the current results
2-add radial density profiles for weak and strong interactions and comment on how condensate is distorted.

Recommendation

Ask for major revision

  • validity: high
  • significance: high
  • originality: high
  • clarity: high
  • formatting: excellent
  • grammar: excellent

Author:  Michał Suchorowski  on 2024-11-26  [id 4990]

(in reply to Report 2 on 2024-09-30)

Dear Referee,

We thank you for your comments. In addition to the comments provided with the resubmitted manuscript, we address them in more detail with supporting figures in the attached file. We hope that the revised manuscript is ready for publication.

Sincerely,
The Authors

Attachment:

Reply__Report_2.pdf

---

## Round 2 · Referee Report · Anonymous (Referee 1) · 2024-12-17

Report

Dear Editor,

The authors addressed all my comments and recommendations from the previous report.
I think the revised manuscript can be accepted for publication.

Sincerely,
Referee 1

Recommendation

Publish (easily meets expectations and criteria for this Journal; among top 50%)

---

## Round 2 · Referee Report · Anonymous (Referee 2) · 2025-1-28

Report

I am happy with the author's response and the changes made to the submission. I will recommend this manuscript for publication.

Recommendation

Publish (meets expectations and criteria for this Journal)

---

## Round 2 · Author Response

Dear Editor,

We appreciate your handling of our submission and the opportunity to revise and resubmit the manuscript.

We are grateful to the Referees for their time and thoughtful review of our paper. We are pleased to note their overall positive evaluation.

The referees' comments helped us improve our manuscript further. For clarity, we have included their original remarks in the reply. In the resubmission, we reply to the referees' comments and provide a list of changes. Moreover, we provide an extended version of our comments, including supporting figures directly as a reply to the reports. In the revised manuscript, major changes are highlighted in blue, while minor corrections (e.g., typos) are not marked. We hope that this revised version meets the standards for publication in SciPost Physics.

Sincerely,
The Authors

---

## Round 2 · List of Changes

Reply to Report #1

Referee 1:
"I think the journal acceptance criteria are fully met."

Our reply:
We thank the Referee for this encouraging evaluation of our work. Below, we address the `Requested changes' of the Referee.

Referee 1:
"1. The physical origin of the impurity-boson interaction is unclear. The manuscript suggests the rotor model for the impurity. Are bosons also rotors?"

Our reply:
We agree that the previous version of the manuscript did not state that bath bosons are point-like particles. Moreover, the motivation behind the choice of the impurity-boson interactions was not included. The revised version contains this information.

Changes to the manuscript:
- We changed the beginning of the first paragraph of section 2.
- We changed the last paragraph in section 2.1 and added a relevant citation to Quantum Theory of Angular Momentum by Varshalovich D. A. et al.

Referee 1:
"2. On p.~4 we read: ``The normalized function $\Psi$ defines the probability of finding a boson at a given position in a molecular frame of reference. It does not depend explicitly on the angle $\varphi_I$...'', however, coordinates $r_i$ explicitly suggest such dependence. I think this issue should be clarified in the revised manuscript."

Our reply:
Following the recommendation of the Referee, we clarified this sentence. In particular, the revised manuscript explains that $\psi$ and function $f$ depend on $\varphi_\text{I}$ only implicitly.

Changes to the manuscript:
- We moved the first sentence (``The normalized function $\psi$ defines the probability of finding a boson at a given position in a molecular frame of reference.'') to the first paragraph of Sec. 2.2.
- We changed the beginning of the second paragraph of Sec. 2.2.

Referee 1:
"3. What do the authors mean by the following sentence: ''Indeed, a strong deformation of the density of the Bose gas requires a number of phonons in momentum space for its description.''?"

Our reply:
We agree that this statement might be unclear. We meant that to describe a strong interaction regime in the momentum space, one would have to include a large number of phonon excitations, which is not trivial.

Changes to the manuscript:
- We changed paragraph 2 of section 4.1.

Referee 1:
"4. The breakdown of the angulon behavior at large angular momenta (see Fig.~6) is signaled as a deviation of the rotation energy from the quadratic law. However, at large $L$ the energy $\Delta E^\text{num}_L$ seems to support a linear dependence. Is it really linear, and is there some physical reasoning for it?"

Our reply:
This is truly an interesting question. In the current version of the manuscript (see the fourth paragraph of Sec. 4.2) we only speculate that the excitation energy follows $\hbar \omega L$. We motivate it by considering a collective excitation of the non-interacting gas. In the outlook (see Sec. 5.2) we mention that it is our plan to investigate these excitations in more detail. We are working on this question at the moment, and therefore we do not suggest any further modifications to the present manuscript.
* * *
Reply to Report \#2

Referee 2:
"This paper is well-organized and clearly written. While some of the results are quite technical, the authors explain them in simple terms. The description of the angulon quasiparticle in real space in terms of the Gross-Pitaevskii equation (GPe) is new and potentially very useful, given that it allows one to capture the physics of non-uniform condensates in the presence of the strong interactions between the boson and the impurity. "

Our reply:
We thank the Referee for a positive evaluation of our work. In the reply, we address the comments of the Referee. We hope that the revised manuscript is ready for publication.

Referee 2:
"The other strength of the manuscript is that the authors consider the system in a harmonic trap so that it can be realized in the experiment, however, this strength also turns out to be its major weakness, because the effects of the trap on the angulon physics are somewhat obscure. In particular, the authors consider the regime where the central density in the trap remains fixed, so for a small number of particles in the system, the potential has to be more confining, and the effects of the trap are important. At the same time, the effects of the trap should be less important as $N \to \infty$. The main analytical result is Eq.(18) and it is derived under the assumption that $N$ is finite, but the trap is turned off. This analytical result agrees well with numerics for a range of $N \gg 1$, but then suddenly breaks down when $N$ becomes larger than some $N^*$. The fact that such behavior does not seem to follow from the formalism used (from what I understand the result in Eq.(18) should hold for arbitrary $N \gg 1$), makes the whole approach look questionable. Since it is not clear what triggers such a behavior, it is not clear whether the harmonic trap has to do with it.

Just to sum up the above, when the authors talk about different regimes based on the number of particles in the system, it is hard to tell whether this is something inherent to the angulon problem, or some of the regimes are the artifacts of the harmonic confinement.

I think to make the discussion clearer and make the claims of the authors stronger, they should also consider the problem in the absence of harmonic confinement, for example, by putting the system into a box of finite size. Then one can fix the density and study the problem as a function of $N$ and the box size $L$. If the results in this system agree with the ones in the current manuscript in the regime where $N \gg 1$, then I will accept the validity of the presented results and will be happy to recommend this manuscript to be published in SciPost. Add results/discussion of the angulon physics in the absence of the harmonic confinement and compare with the current results."

Our reply:
We thank the Referee for this suggestion. We have performed calculations in a box potential as suggested in the report, see Fig. S1 in Reply to the Report. The plot shows the angular momentum of the bath $A$ as a function of the size of the system in the units of the healing length, $\xi$. Solid lines show $A$ for a system in a harmonic trap (where $R$ denotes Thomas-Fermi radius), dashed lines show values for the system in a hard wall potential (where $R$ denotes the radius of the potential), and dotted horizontal lines show analytical results for the angulon state. It is clear that while the departure point from the angulon plateau is shifted, the collective excitations in the bath dominate the physics in the limit $N\gg 1$ for both potentials. In fact, the departure from the angulon plateau is more dramatic for a box potential.

Note that in Sec. 3.3, we discuss that the collective excitations of the condensate are expected to have energy lower than the angulon state. Our theoretical formalism (Eq. (18)) does not take these excitations into account, which is the reason why our numerical results agree with theoretical predictions only in the regime where the angulon state is a ground state. Even though collective excitations will appear in any confining potential, the shape of the confinement will affect the exact energy dispersion (and, in effect, the departure point from the angulon plateau).

In the revised version, we added a few sentences to strengthen our discussion. However, we do not add Fig. S1 to the manuscript, as it is already very long in our opinion. Of course, if the Referee believes that this figure must be necessary included, we will reconsider our decision.

Changes to the manuscript:
- We added relevant comment at the end of paragraph 5 of Sec. 3.3
- We change the beginning of the paragraph 2 in section 5.2

Referee 2:
Finally, the angulon problem is a close cousin of the Bose polaron problem, as was also pointed out by the authors. For the Bose polarons, there is a thermodynamic relation between the number of particles inside the polaronic cloud and its energy, see https://doi.org/10.1103/PhysRevLett.126.123403. Can one expect something similar to hold for angulons?

Our reply:
We thank the Referee for this comment, it is well taken. We agree that it is interesting to see if a similar thermodynamic relation is relevant to the angulon problem. Based on our theoretical framework, we established a relation similar to the one in the cited paper, see the new discussion in App. D. However, further work is needed to define the number of particles inside the angulon cloud, which we leave for further studies.

Changes to the manuscript:
- We added relevant paragraphs in Sec. 3.2 and App. D.

Referee 2:
Add radial density profiles for weak and strong interactions and comment on how condensate is distorted.

Our reply:
Please note that Fig.~10 in App.~D shows the densities obtained numerically and analytically in the angulon regime. We believe that these plots give more information than the radial density profiles as the problem is two-dimensional, and angular dependence is crucial to understanding the properties of the system. However, if the Referee believes that such plots would do the paper good, we will include them. Figure S2 in Reply to the Report shows samples of density profiles.

---

## Editorial Decision

published